# RLAnything: Forge Environment, Policy, and Reward Model in Completely Dynamic RL System

**Yinjie Wang** [1]   **Tianbao Xie**   **Ke Shen**   **Mengdi Wang** [2]   **Ling Yang** [2]

**Project:** https://github.com/Gen-Verse/Open-AgentRL

## Abstract

The quality of both the environment and the reward model fundamentally governs the effectiveness of reinforcement learning. Accordingly, we propose **RLAnything**, a reinforcement learning framework that dynamically optimizes each component through closed-loop optimization, amplifying learning signals and strengthening the overall system. Specifically, the policy is trained with integrated feedback from step-wise and outcome signals, while the reward model is jointly optimized via consistency feedback, which in turn further improves policy training. Moreover, our theory-motivated automatic environment adaptation improves training for both the reward and policy models by leveraging critic feedback from each, enabling learning from experience. Empirically, each added component consistently improves the overall system, and RLAnything yields substantial gains in practical applications, boosting Qwen3-VL-8B-Thinking by $8.5\%$ on OS-World and Qwen2.5-7B-Instruct by $21.2\%$ and $12.1\%$ on AlfWorld and LiveBench, respectively.

## 1. Introduction

Reinforcement learning with verifiable rewards (RLVR) is an effective approach for improving the reasoning capabilities of large language models (OpenAI, 2024; Shao et al., 2024; Guo et al., 2025a). However, as real-world applications extend beyond single-turn question answering, especially when policies interact with environments iteratively over long trajectories, binary outcome rewards alone provide insufficient supervision (Xiong et al., 2024; Lightman

et al., 2023; Xi et al., 2025). Step-wise signals are typically provided by generative reward models, which often outperform scalar-based models by leveraging the reasoning capabilities of language models (Zhang et al., 2024; Liu et al., 2025). However, training these models usually requires collecting high-quality, task-specific supervision (Xi et al., 2025; Zhang et al., 2025), motivating the need for more automated methods and scalable supervision.

Beyond reward design, the quality of the environment is also vital for scaling reinforcement learning. Aligning task difficulty with a model's current capabilities is known to improve training dynamics (Yu et al., 2025; Yang et al., 2025). In RLVR, it has been shown that adapting task difficulty during optimization can improve policy training (Zeng et al., 2025). In real-world environments, such as computers for GUI agents (Xie et al., 2024; Wang et al., 2025a) or the physical world for robots (Kober et al., 2013), the scope of exploration is largely defined by the task. Moreover, scaling the environment by increasing task diversity can further promote policy generalization in broader scenarios (Cobbe et al., 2020; Team et al., 2021; Fang et al., 2025; Cai et al., 2025; Song et al., 2026; Chen et al., 2025).

If there exists an RL system that jointly optimizes the environment, policy, and reward model to amplify learning signals and strengthen the overall system?

In this work, we propose **RLAnything**, a dynamic RL framework that forges the environment, policy, and reward model in a closed-loop system, where each component continuously receives feedback from the others to amplify learning signals. First, the policy is trained with integrated feedback that combines verifiable outcome rewards with step-wise signals provided by the reward model. Second, the reward model is jointly optimized via consistency feedback based on outcome and self-consistency, producing reliable step-wise supervision that in turn improves policy learning. Finally, motivated by our theoretical results, we show that balancing task difficulty benefits not only policy training but also reward model training in our RL system. Accordingly, we adapt environment tasks using critic feedback from both the policy and reward model, enabling precise

[1]University of Chicago [2]Princeton University. Correspondence to: Ling Yang <yangling0818@163.com>, Mengdi Wang <mengdiw@princeton.edu>.

*Proceedings of the 43rd International Conference on Machine Learning*, Seoul, South Korea. PMLR 306, 2026. Copyright 2026 by the author(s).

*Figure 1.* Motivation and takeaways of our RLAnything framework. First, in complex real-world applications, reinforcement learning benefits from integrating step-wise rewards with outcome rewards. Second, the reward model can be jointly optimized with the policy via outcome supervision and self-consistency signals. Third, we show that adapting environment task difficulty to the policy's capability not only facilitates policy learning but also improves reward model training within our framework. Environment tasks leverage critic feedback from both the policy and the reward model to drive automatic, targeted adaptation, further enabling active learning from experience.

and automatic task adjustment. In particular, we feed the reward model's summarized information, which captures the policy's failures, into a language model to perturb the task, providing concrete guidance on how to modify it. To demonstrate the generality of our framework, we conduct empirical studies in three representative scenarios on computer use setting (Xie et al., 2024), text-based interactive games (Côté et al., 2018; Shridhar et al., 2020), and coding LLMs. We summarize our main contributions as follows:

- We propose RLAnything, a fully dynamic RL system that forges the environment, policy, and reward model through closed-loop optimization to amplify learning signals and strengthen the overall system, guided by our theoretical insights.

- Across computer-use agents, text-based LLM agents, and coding LLMs, we show that each added dynamic component consistently benefits the overall system and improves out-of-distribution performance.

- We achieve significant gains in practical applications: Qwen3-VL-8B-Thinking improves by $8.5\%$ on OS-World, and Qwen2.5-7B-Instruct improves by $21.2\%$ and $12.1\%$ on AlfWorld and LiveBench, respectively.

- We show broad applicability: optimized reward-model signals outperform outcomes that rely on human labels, enabling active learning from experience and potential environment scaling.

## 2. Related Works

### 2.1. Reinforcement Learning of Large Language Models

Reinforcement learning with verifiable rewards has been used to enhance the reasoning abilities of language models (Guo et al., 2025a; Team et al., 2025; Hugging Face, 2025; OpenAI, 2024; Hu et al., 2025; Yu et al., 2025; Wang et al., 2025c) and has been applied across diverse settings, including coding tasks (Yang et al., 2024b; Wang et al., 2025e)

and RAG tasks (Jiang et al., 2025; Jin et al., 2025). With the widespread adoption of agentic AI, RL has also been extended to multi-turn settings (Dong et al., 2025; Wang et al., 2025a; Li et al., 2025b; Lu et al., 2025a;b; Team, 2025; Wang et al., 2025d; Zhou et al., 2025), where the policy model interacts with an environment over long trajectories. However, reward sparsity (Lightman et al., 2023; Xi et al., 2024) and the limited scale of existing environments (Cobbe et al., 2020; Team et al., 2021; Zhou et al., 2023; Yao et al., 2022; Xie et al., 2024) remain key challenges.

### 2.2. Reward Modeling and Environments

Reward models, especially generative reward models, play an important role in making reinforcement learning practical (Zhao et al., 2025; Lightman et al., 2023). In RLVR settings, a single outcome reward can jointly optimize both the reward model and the policy (Wang et al., 2025e; Zha et al., 2025), but this does not directly extend to multi-turn settings due to the lack of step-wise supervision. The quality of the environment tasks is also critical for effective RL (Zhou et al., 2023; Yao et al., 2022; Xie et al., 2024). Prior work has shown that adjusting task difficulty can improve policy training (Yang et al., 2025; Yu et al., 2025), motivating methods that generate or modify tasks to strengthen learning signals. (Zala et al., 2024; Xiong et al., 2025; Guo et al., 2025b). For example, Zeng et al. (2025) builds an RLVR engine where each task comes with multiple difficulty levels; Xue et al. (2026) scales this direction via verifiable, automated task synthesis. However, these systems lack the step-wise signal for long-horizon tasks.

## 3. Method

Our **RLAnything** framework (see Algorithm 1) tightly couples the policy model, reward model, and environment to achieve joint optimization. Specifically, we train the policy using integrated feedback (Equation 1 in Section 3.1), which combines step-wise signals from reward model with

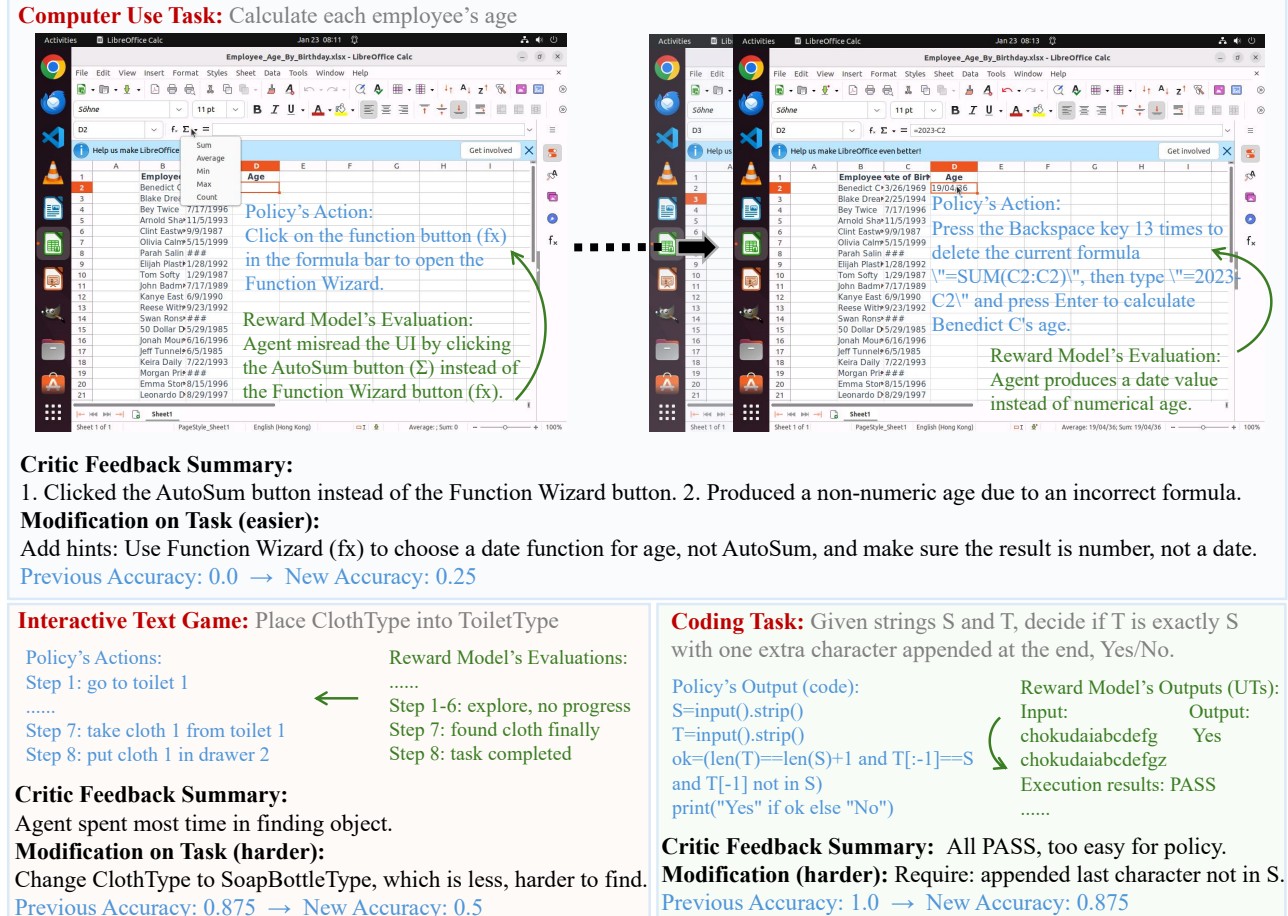

*Figure 2.* Examples of environment task adaptation based on critic feedback across computer use agent, text-game agent, and coding LLM in our experiments. The critic feedback is summarized from the reward model's evaluations and is used to automatically adapt tasks.

the trajectory-level outcome signal. We train the reward model by treating the policy's trajectories as environment tasks and assigning consistency feedback via Equation 2; we will prove that this objective also improves the reward model's accuracy in predicting final outcomes in Section 3.3. As the reward model becomes more accurate, it in turn provides a stronger and more informative learning signal for the policy. Motivated by the theoretical insights in Section 3.3, we show that adapting the difficulty of environment tasks benefits not only policy training but also reward model training. To achieve automatic and targeted adaptation, concrete task modifications are guided by critic feedback summarized from the reward model's evaluative responses (Section 3.4). Overall, RLAnything enables the environment, policy, and reward model to provide feedback to one another, strengthening learning signals and improving the system as a whole.

### 3.1. Integration Feedback for Policy

Given a task $q \in Q$ and a policy $\pi$, we sample a trajectory $\tau \sim \pi(\cdot \mid q)$ and obtain a final outcome reward $O_\tau \in$

$\{-1, 1\}$. For the $i$-th step $\tau_i$, we query a reward model $m$ independent times, yielding $S_{\tau_i, j} \in \{-1, 1\}$ for $j = 1, \ldots, m$, where $-1$ indicates no progress toward the final goal or a step-wise error. We then define the step reward as

$$R_{\tau_i} = O_\tau + \frac{\lambda}{m} \sum_{j=1}^{m} S_{\tau_i, j}, \qquad (1)$$

which combines the outcome signal with nuanced step-wise feedback; we set $\lambda = 1$ by default. Finally, we compute advantages by standardizing rewards across trajectories at the same step index $i$, i.e., over the set $\{R_{\tau_i} : \tau \sim \pi(\cdot \mid q)\}$.

### 3.2. Consistency Feedback for Reward Model

For the $i$-th step of a trajectory, $\tau_i$, the $j$-th evaluation (out of $m$ in total) by the reward model $r_\phi$ assigns a step-level label $S_{\tau_i, j}$ and receives the following reward signal:

$$R_{S_{\tau_i, j}} = R_{\tau_i} \cdot S_{\tau_i, j}. \qquad (2)$$

Intuitively, $R_{\tau_i} \in [-1, 1]$ reflects the overall quality of step $\tau_i$: $R_{\tau_i} < 0$ indicates poor quality, while $R_{\tau_i} > 0$

indicates good quality. Values of $R_{\tau_i}$ closer to $0$ indicate greater uncertainty about this step. The agreement between this step-level signal and the $j$-th evaluation is captured by $R_{\tau_i} \cdot S_{\tau_i,j}$, which we use as the supervision signal for that evaluation. During policy optimization, the policy's trajectories also serve as the training environment for the reward model. Meanwhile, the refined reward model provides a stronger reward signal for the policy and more accurate feedback to guide environment task adaptation, which in turn facilitates training of both the policy and the reward model. In Section 3.3, we prove that optimizing this objective for reward model improves its accuracy in predicting future outcome, and our environment adaptation further facilitates this optimization.

## 3.3. Adaptation of Environment Benefits Both Policy and Reward Models

The quality of the reward model's signal depends not only on the logical correctness of an individual step, but also on its ability to predict that step's future impact. Specifically, we want to optimize the following reward precision

$$\mathcal{A} = P(S_{\tau_i^+} > S_{\tau_i^-} \mid O_{\tau^+} = 1, O_{\tau^-} = -1),$$

where $S_{\tau_i} = \sum_{j=1}^{m} S_{\tau_i,j}/m$ is the mean process reward assigned by $r_\phi$. In the following theorem, we show that this reward precision can be translated into an objective that can be approximated by our reward design: $\mu \triangleq p_+ + p_-$, where $p_+ = P(S_{\tau_i^+,j} = 1)$ and $p- = P(S_{\tau_i^-,j} = -1)$.

**Theorem 3.1.** $\mathcal{A} \to 1$ as $m \to \infty$ if and only if $\mu > 1$. Moreover, $\mathcal{A} \geq 1 - e^{-m(\mu-1)^2/4}$ when $\mu > 1$.

The target $\mu = p_+ + p_-$ indicates that the sampling densities used to estimate $p_+$ and $p_-$ should be balanced rather than heavily skewed toward one side; otherwise, the estimator can be dominated by a single class, leading to biased evaluation. However, this balance can break when the training environment for the reward model, induced by the policy's trajectories, is not balanced in task difficulty. We use the following theorem to formalize this result.

**Theorem 3.2.** *The left-hand side is the RL objective for the reward model. When $\lambda = 1$, on the right-hand side, $f_+ \geq 0$ and $f_- \geq 0$ are importance-weight functions of $\tau$, and $\langle \cdot, \cdot \rangle$ denotes the $L^2$ inner product over $\tau \sim \pi_\theta(\cdot \mid q)$.*

$$\mathbb{E}_{\substack{q \sim Q \\ \tau \sim \pi_\theta(\cdot|q)}} \mathbb{E}_{S_{\tau_i,j} \sim r_\phi(\cdot|\tau_i)} \left[ R_{S_{\tau_i,j}} \right]$$
$$= 4 \mathbb{E}_{q \sim Q} \left[ \langle p_+, f_+ \rangle + \langle p_-, f_- \rangle \right] + C,$$

*where the importance-weight norm ratio $\|f_+\|/\|f_-\| \to 0$ as $P(O_\tau = -1 \mid q, \pi_\theta) \to 1$, and $\|f_+\|/\|f_-\| \to \infty$ as $P(O_\tau = 1 \mid q, \pi_\theta) \to 1$. $C$ is constant irrelevant to $\phi$, and $\|\cdot\|$ is $L^2$ norm.*

---

**Algorithm 1** RLAnything Pipeline Overview
___
1: **Given:** environment task set $Q$; policy $\pi_\theta$; reward model $r_\phi$; thresholds $\alpha_{\text{high}} = 0.8, \alpha_{\text{low}} = 0.2$.
2: **for** $k = 1, \dots, K$ **do**
3:    **Sampling:**
4:    Sample task $q \sim Q$.
5:    $\pi_\theta$ samples trajectories $\mathbb{T}_q$, each $\tau \in \mathbb{T}_q$, $\tau = (\tau_1, \dots, \tau_T)$. Outcome $O_\tau \in \{-1, 1\}$.
6:    Reward model samples reasoning $r_{\tau_i,j}$ and final score $S_{\tau_i,j} \in \{-1, 1\}$ for $\tau_i, 1 \leq j \leq m$.
7:    Compute step-wise quality for each step $\tau_i, R_{\tau_i} = O_\tau + \lambda \sum_{j=1}^{m} S_{\tau_i,j}/m$, for $\lambda > 0, i = 1, \dots, T$.
8:    **Update policy $\pi_\theta$:**
9:    Compute advantages $A_{\tau_i}^{\pi_\theta}$ by standardize $R_{\tau_i}$ across $\tau \in \mathbb{T}_q$ at same $i$, for each $q$.
10:    Train $\pi_\theta$ with $A_{\tau_i}^{\pi_\theta}$ for action step $\tau_i$.
11:    **Update reward model $r_\phi$:**
12:    Compute advantages $A_{\tau_i,j}^{r_\phi}$ by standardize $R_{\tau_i} \cdot S_{\tau_i,j}$ across $j$, for each $\tau_i$.
13:    Train $r_\phi$ with $A_{\tau_i,j}^{r_\phi}$ for reward reasoning $r_{\tau_i,j}$.
14:    **Adapt environment task:**
15:    $s \leftarrow$ summarize step-wise errors $(\{(\tau_i, r_{\tau_i,j}\}_{i=1}^{T})$.
16:    **if** $\text{acc}(q) > \alpha_{\text{high}}$:
17:      Propose harder task: $q' \leftarrow \text{harder}(q; s)$.
18:      **if** $\alpha_{\text{low}} < \text{acc}(q') < \text{acc}(q)$: replace $q \leftarrow q'$
19:    **else if** $\text{acc}(q) < \alpha_{\text{low}}$:
20:      Propose easier task: $q' \leftarrow \text{easier}(q; s)$.
21:      **if** $\text{acc}(q) < \text{acc}(q') < \alpha_{\text{high}}$: replace $q \leftarrow q'$
22: **end for**

---

This demonstrates that when a task $q$ is overly difficult, i.e., $P(O_\tau = -1 \mid q, \pi_\theta) \to 1$, or overly easy for the policy model, i.e., $P(O_\tau = 1 \mid q, \pi_\theta) \to 1$, importance sampling becomes extremely unbalanced between $p_+$ and $p_-$, violating the reward-precision target $p_+ + p_-$ established in Theorem 3.1. Given this insight, in our reward system, moderating the difficulty of task $q$ can not only facilitate policy training but also improve training of the process reward model. We will introduce how to adapt task automatically in the following Section 3.4.

## 3.4. Critic Feedback for Environment Tasks

In our framework, we estimate task difficulty using the policy's rollout accuracy. When the accuracy falls outside predefined thresholds ($\alpha_{\text{low}}$ and $\alpha_{\text{high}}$), we prompt a language model to modify the task to make it harder or easier while preserving the essence of the original task (see prompts in Appendix C.7). The concrete modifications are guided by summarized evaluative traces from the reward model, $r_{\tau_i,j}$, which are generated when obtaining $S_{\tau_i,j}$. We summarize only steps $\tau_i$ that exhibit potential failures, namely those for which $S_{\tau_i,j} = -1$ for some $j$, to capture the

*Table 1.* Each dynamic component consistently improves policy and reward model optimization across GUI agent, LLM agent, and coding LLM settings. "Policy + Reward + Env" denotes RLAnything, which keeps policy, reward model, and environment dynamic during training. "Policy + Reward" jointly optimize reward model but lacks environment adaptation. "Policy" uses integrated feedback with a fixed reward model and environment. "In Domain" denotes accuracy on in-domain tasks, while "OOD" denotes accuracy on out-of-distribution tasks. "Process Acc" denotes the accuracy of evaluating step-wise correctness, while "Outcome Acc" denotes the accuracy of predicting outcomes using process rewards. We evaluate on three datasets in coding setting. "Code" denotes coding accuracy. "UT" denotes the accuracy of generated unit tests. "Detect" denotes the accuracy of unit tests in identifying code correctness.

| Dynamic Components | GUI Agent | | | | LLM Agent | | | |
| --- | --- | --- | --- | --- | --- | --- | --- | --- |
| | Policy Acc | | Reward Acc | | Policy Acc | | Reward Acc | |
| | In Domain | OOD | Process | Outcome | In Domain | OOD | Process | Outcome |
| Before Optimization | 30.0 | 22.4 | 86.0 | 55.2 | 39.0 | 44.9 | 47.0 | 59.6 |
| Policy | 35.3 | 24.8 | 86.0 | 55.2 | 51.1 | 59.3 | 47.0 | 59.6 |
| Policy + Reward | 36.9 | 26.0 | 88.7 | 57.6 | 55.6 | 61.8 | 55.2 | 61.1 |
| Policy + Reward + Env | **40.6**$_{+10.6}$ | **27.4** $_{+5.0}$ | **91.3** $_{+5.3}$ | **60.4** $_{+5.2}$ | **60.2**$_{+21.2}$ | **63.6**$_{+18.7}$ | **59.4**$_{+12.4}$ | **61.7** $_{+2.1}$ |

| Dynamic Components | LiveBench | | | LiveCodeBench | | | CodeContests | | |
| --- | --- | --- | --- | --- | --- | --- | --- | --- | --- |
| | Code | UT | Detect | Code | UT | Detect | Code | UT | Detect |
| Before Optimization | 31.1 | 27.8 | 19.6 | 26.9 | 35.7 | 28.1 | 21.2 | 43.8 | 35.7 |
| Policy | 38.8 | 27.8 | 19.6 | 31.4 | 35.7 | 28.1 | 25.8 | 43.8 | 35.7 |
| Policy + Reward | 40.0 | 73.3 | 37.9 | 31.7 | 81.4 | 49.0 | 27.5 | 86.0 | 55.4 |
| Policy + Reward + Env | **43.2**$_{+12.1}$ | **78.9**$_{+51.1}$ | **48.5**$_{+28.9}$ | **34.1** $_{+7.2}$ | **81.6**$_{+45.9}$ | **55.2**$_{+27.1}$ | **28.3** $_{+7.1}$ | **87.1**$_{+43.3}$ | **67.9**$_{+32.2}$ |

policy's likely error patterns (Appendix C.6). Task adaptation therefore hinges on accurate critic feedback and, in return, yields more effective adaptation that benefits both the policy and the reward model. We also enforce quality control for modified tasks. If the goal is to make the original task $q$ harder, we accept the modified task $q'$ only if $\alpha_{\text{low}} < \text{acc}(q') < \text{acc}(q)$; if the goal is to make it easier, we accept $q'$ only when $\text{acc}(q) < \text{acc}(q') < \alpha_{\text{high}}$. This helps ensure the validity of the new task and the effectiveness of adaptation (Algorithm 1). We then replace the original task with the accepted task $q'$ in the task set $Q$.

# 4. Experiments

In this section, we demonstrate our takeaways (Figure 1) and evaluate the performance of the optimized models through extensive experiments. Specifically, we focus on two real-world agentic settings: computer-use agents and text-based interactive games, where large language models serve as both the policy and reward models, and we perform automatic environment adaptation. In addition, we also demonstrate the effectiveness of our framework for RLVR coding tasks, where no interactive environment is available.

## 4.1. Experiment Settings

### 4.1.1. MODELS AND OPTIMIZATIONS

For GUI agents on OSWorld (Xie et al., 2024), we use Qwen3-VL-8B-Thinking as both the policy and reward model. We set the maximum number of interaction steps to 50 for evaluation and 30 for RL rollouts. At each RL step, we sample 12 tasks, each with 8 independent rollout trajectories. For the reward model, we perform 3 evaluations per

policy response. We use Qwen3-4B (Yang et al., 2025) for task adaptation. For the LLM agent on AlfWorld (Shridhar et al., 2020; Côté et al., 2018), we use Qwen2.5-7B-Instruct as the policy model and Qwen2.5-14B-Instruct as the reward model (Yang et al., 2024a), and we use Qwen3-4B (Yang et al., 2025) for task adaptation. We set the maximum number of steps to 60 for evaluation and 40 for RL rollouts. At each RL step, we sample 16 tasks, each with 8 independent rollouts. For coding LLMs, we use the same model combination as in the AlfWorld setting. At each RL step, we sample 64 tasks, with 32 independent code-solution generations and 32 independent unit-test generations per task. We evaluate on LiveCodeBench-V2 (Jain et al., 2024), CodeContests (Li et al., 2022), and LiveBench (White et al., 2024), and we train on CodeContests (Li et al., 2022). Additional experimental details are provided in Appendix C.

### 4.1.2. REWARD MODELING AND ENVIRONMENT

For reward modeling, we use an LLM as a generative reward model: it is prompted to assess each step's quality and its potential impact on the final outcome, and then outputs 1 or −1 after reasoning (Appendix C.5). For the GUI agent, we provide as context a summary of previous actions, the two most recent images, and the action to be evaluated between them. For AlfWorld, we summarize prior actions and their observed consequences. In the coding-LLM setting, we use a unit-test generator as the reward model, where each newly generated test evaluates one aspect of the code.

For environment adaptation, we first summarize the reward model's outputs for steps flagged as potentially erroneous (Appendix C.6), and then feed this critic feedback to a language model to rewrite the task toward a target perturbation

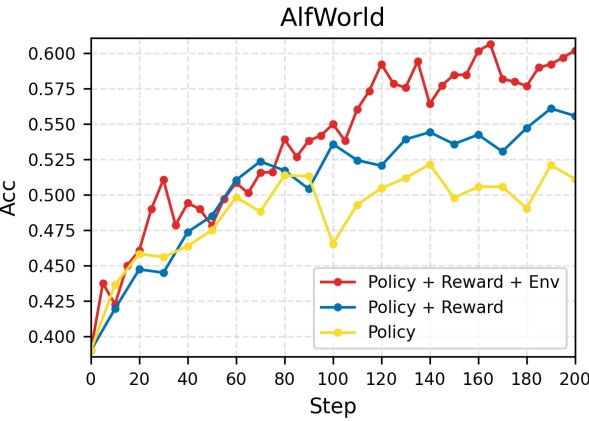 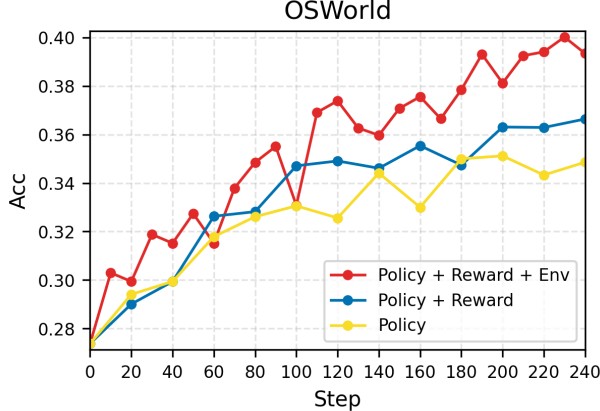

*Figure 3.* Each dynamic component consistently improves policy's training curve across LLM agent and GUI agent settings.

(easier or harder) (Appendix C.7). In the coding-LLM setting, the critic feedback is simply the code's evaluation results on the generated unit tests.

## 4.2. RLAnything Facilitates Policy Training

To show that each dynamic component consistently improves policy optimization, we report three training curves in Figure 3, where each method adds one additional dynamic component. For intermediate evaluations along the curves, we set the maximum interaction steps to 30 for OS-World and 60 for AlfWorld. We find that making both the reward model and the environment dynamic yields stronger optimization and a higher convergence point. Specifically, jointly optimizing the reward model improves the supervision signal, which in turn benefits policy training. Moreover, environment adaptation benefits not only the policy but also the reward model (Section 4.3), thereby yielding a three-fold gain for policy training. We also report final evaluations on in-domain and out-of-distribution (OOD) tasks (see details in Appendix C.2) in Table 1. The substantial improvements on OOD tasks highlight the stronger generalization of our optimization framework.

## 4.3. RLAnything Produces a Stronger Reward Model

From Table 1, we draw two conclusions. First, our reward design (Equation 2) effectively improves the reward model. Second, adapting environment tasks further facilitates reward model training, supporting our theoretical results. To evaluate improvements in the reward model, we consider two aspects: (i) its ability to assess step-wise quality (*process accuracy*), and (ii) its ability to predict a step's influence on the final outcome (*outcome accuracy*). The ground truth for outcome accuracy comes from verifiable outcomes, while step-wise quality labels are obtained by majority voting from a stronger reasoning model prompted to assess step-level quality; details are provided in Appendix C.3.

From Table 1, we find that both accuracy metrics improve across all settings after optimization, and that environment adaptation further enhances them. We also conduct ablation studies using different supervision models in Appendix B.2, which show similar results.

## 4.4. Adaptation of Environments Enables Active Learning from Experience

Our environment adaptation is automated and explicitly guided by critic feedback from the reward model, which diagnose the policy's likely errors on a given task. In this section, we provide examples showing how this targeted adaptation promotes more active policy learning. In the example in Figure 2, the GUI agent fails to obtain any successful trajectories across independent rollouts. The reward model pinpoints two specific mistakes made by the policy on this task, and its outputs serve as diagnostic feedback for rewriting the task. The adapted prompt adds targeted tips, enabling the policy to achieve successful rollouts and learn more effectively, rather than relying on random exploration. Moreover, tasks can also be adapted in the opposite direction to encourage more challenging exploration. In the interactive text-game example, the policy succeeds across trajectories but spends most steps searching for the object; the model therefore increases difficulty by replacing the target object with another that appears less frequently. See additional examples in Appendix B.5.

## 4.5. State-of-the-Art Performance of the Optimized Multimodal GUI Agent

We compare our RLAnything-optimized GUI agent with strong open-source baselines, including UI-TARS1.5-7B (Qin et al., 2025), OpenCUA-7B (Wang et al., 2025b), and Qwen3-VL-8B-Thinking (Bai et al., 2025). As shown in Figure 4, our method achieves the best performance across all OSWorld task categories, highlighting the effectiveness

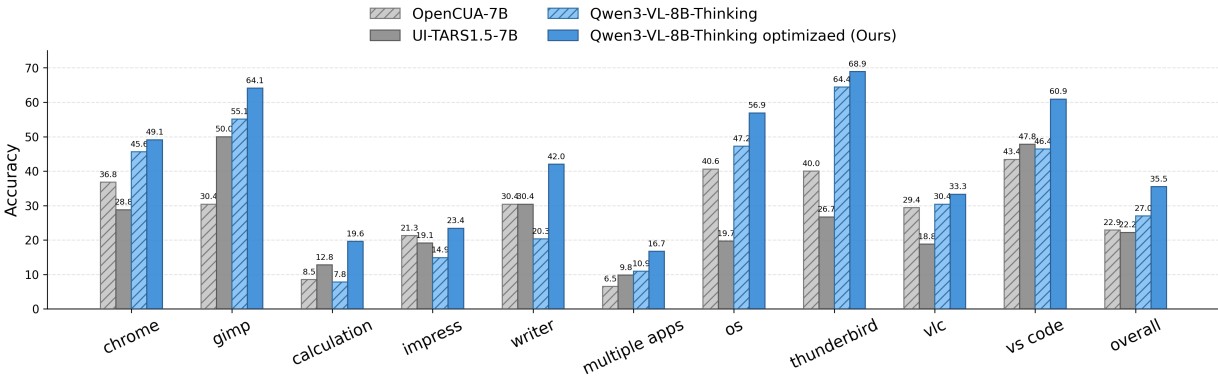

*Figure 4.* Results on OSWorld tasks for different models, including UI-TARS1.5-7B (Qin et al., 2025), OpenCUA-7B (Wang et al., 2025b), Qwen3-VL-8B-Thinking (Bai et al., 2025), and our optimized model. Results are averaged over three independent runs, with the maximum number of interaction steps set to 50.

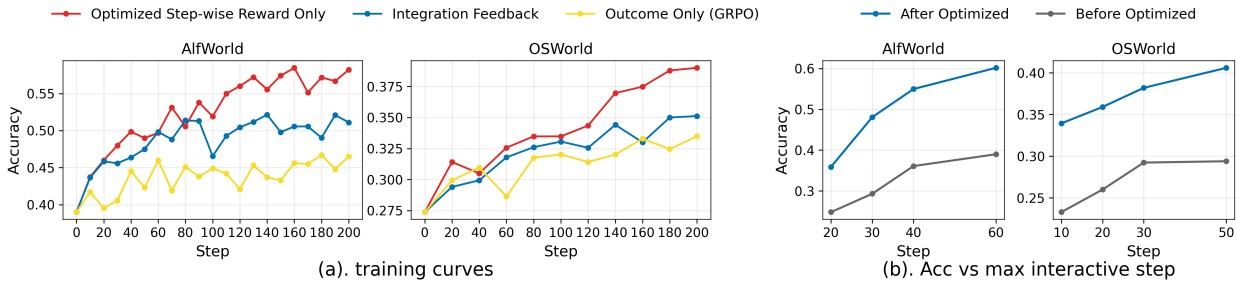

*Figure 5.* (a) shows the need for an integrated reward and that our optimized reward model alone provides a stronger learning signal than outcome supervision (the standard GRPO (Shao et al., 2024) setting). (b) shows scaling with number of interaction steps.

of our optimization framework. Notably, our optimized model improves accuracy by $8.5\%$ on OSWorld. On out-of-distribution tasks, the model also improves by $5.0\%$.

### 4.6. Advantages of Integrating Step-wise and Outcome Rewards for Policy Training

In complex real-world settings where policies must interact with environments over long trajectories to achieve sufficient exploration (see Figure 5(b)), outcome rewards are too sparse to provide an effective training signal. We compare the commonly used outcome-only reward with our integrated reward design (Equation 1) using the RL training curves in Figure 5 (a) for both the LLM agent and GUI agent settings. The results highlight the necessity of integrated rewards, which combine nuanced step-wise signals with faithful supervision from verifiable final outcomes.

### 4.7. Optimized Reward Model Supervision Outperforms Human-Labeled Outcome Supervision

In complex real-world environments such as computer-use tasks, defining verifiable outcomes often requires human effort. In particular, GUI evaluators are typically implemented as human-written evaluation scripts, which limits environ-

ment scaling for exploration and training. We propose using only the step-wise signals provided by our optimized reward model, which can evaluate both the current action and its future influence. Specifically, we train the policy using only our optimized reward model for step-wise supervision, without any outcome rewards from evaluator scripts. Surprisingly, this setting even outperforms training with verifiable outcome rewards (see Figure 5 (a)), demonstrating the effectiveness of our framework in improving reward models and its potential to enable large-scale, self-evolving agents in real-world environments such as computers.

### 4.8. Also Works for Single-Turn Coding Tasks

Beyond interactive settings, RLAnything also applies to RLVR-style coding tasks: policy reward is the code pass rate over all unit tests (dataset-provided plus tests generated by a unit-test generator that serves as the reward model). We assign rewards to each generated unit test (UT) as follows. We label a generated code as *gt code* if it passes all dataset-provided gt UTs, and label a generated UT as a *gt UT* if it passes on all gt codes. A gt UT receives reward equal to the number of non-gt codes that it causes to fail; otherwise, it receives the negative number of non-gt codes that it incorrectly lets pass. We show this is equivalent to

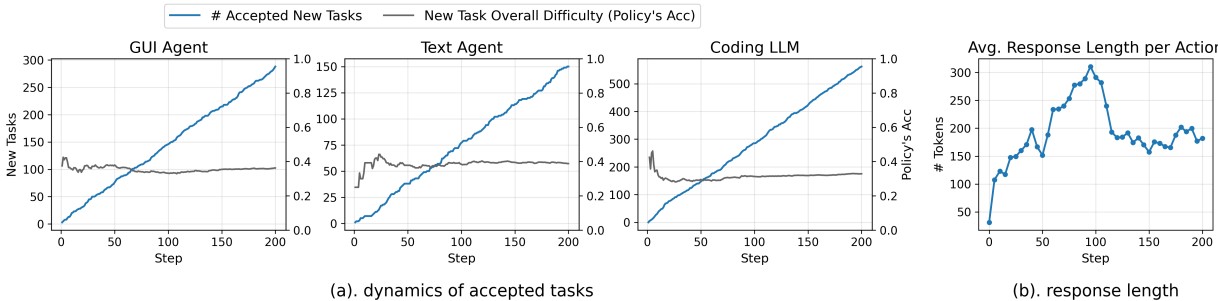

*Figure 6.* (a). The dynamics of accepted new tasks across three settings, including the number of accepted new tasks and the policy's accuracy on these tasks over steps. (b). The average response length per action step during AlfWorld training.

our general framework in Appendix A.2. To evaluate the reward model, we measure both the correctness of generated UTs and their accuracy in detecting code correctness (Appendix C.3); overall, Table 1 shows that both coding performance and unit-test generation quality improve through joint UT training and environment adaptation.

### 4.9. Trade-off Between Outcome and Self-consistency Supervision in Optimization

In our integrated reward design $R_{\tau_i}$, supervision from the final outcome $O_\tau$ and the aggregated step-wise signal $\lambda \sum_{j=1}^{m} S_{\tau_i,j}/m$ are balanced by the hyperparameter $\lambda$. Beyond its effect on the policy, $\lambda$ also influences reward-model supervision: larger $\lambda$ places more emphasis on step-level quality and less on predicting outcome influence. We conduct an ablation on $\lambda$ in the AlfWorld setting over 100 RL training steps, evaluating both the policy and the reward model. Specifically, to study the effect on the reward model, we fix $\lambda = 1$ in $R_{\tau_i}$ and vary $\lambda$ in $R_{S_{\tau_i,j}}$; to study the effect on the policy, we fix $\lambda = 1$ in $R_{S_{\tau_i,j}}$ and vary $\lambda$ in $R_{\tau_i}$. As shown in Table 2, $\lambda$ indeed trades off outcome- and self-consistency-based supervision, and policy optimization performs best at $\lambda = 1$, which we use by default. Reported numbers are averaged over the last three evaluations on the training curve, each averaged over three independent runs. We also discuss the influence of different choices of $\lambda$ on theoretical results in Appendix A.2.

*Table 2.* Influence of $\lambda$ on optimization for policy and reward model in RLAnything framework.

|  | $\lambda$ | 1 / 4 | 1 | 4 |
|---|---|---|---|---|
| Reward Model | process acc | 51.8 | 54.7 | 55.5 |
|  | outcome acc | 62.5 | 61.0 | 60.2 |
| Policy Model | acc | 48.0 | 54.1 | 53.3 |

### 4.10. Dynamics of Accepted New Tasks

In this section, we analyze the tasks accepted during optimization (see Figure 6 (a)). First, the number of accepted

tasks grows approximately linearly with training steps, indicating the potential for environment scaling. We then characterize task difficulty using the policy's accuracy on these accepted tasks. Accuracy fluctuates early due to limited samples but quickly stabilizes at a moderate level; the converged value is below $0.5$ because the original tasks are mostly challenging for the policy. Finally, we assess the quality of accepted tasks using a much stronger reasoning model by running 16 independent trials per task and reporting the rate of at least one successful run. We use Qwen3-VL-32B-Thinking for the GUI setting and Qwen3-32B for AlfWorld and the coding setting, obtaining pass-at-least-one rates of $96.0\%$, $96.7\%$, and $94.2\%$, respectively. These results demonstrate the effectiveness of our acceptance mechanism in filtering out incorrect synthetic tasks.

### 4.11. Response Length on AlfWorld

We study response length and reasoning patterns in the AlfWorld setting (Figure 6(b)). Initially, the policy model (Qwen2.5-7B-Instruct) often fails to produce adequate reasoning before taking actions. After optimization, its chain-of-thought length increases rapidly, and by the end of training, responses become more stable and efficient while still maintaining sufficient reasoning ability.

## 5. Conclusion

We introduce RLAnything, a dynamic closed-loop RL framework that jointly optimizes the environment, policy, and reward model to amplify learning signals. RLAnything trains the policy with integrated supervision and improves the reward model via consistency feedback, providing stronger and more reliable signals throughout training. Motivated by our theory, we show that balancing task difficulty benefits both policy and reward optimization, and implement automatic environment adaptation using critic feedback from both. Experiments across GUI agent, LLM agent, and coding LLM settings verify RLAnything's effectiveness, yielding substantial gains on benchmarks such as OSWorld, AlfWorld and LiveBench.

## Impact Statement

This paper advances reinforcement learning for large language models by proposing a closed-loop framework that co-optimizes the policy, reward model, and environment to provide stronger, more reliable learning signals. Our results in real-world GUI settings suggest a practical path toward self-evolving agents in digital worlds, where both task difficulty and supervision adapt to the agent's capability. At the same time, improving autonomous computer-use abilities amplifies misuse risks, underscoring the need for verifiable tasks, auditable feedback, and deployment safeguards such as access control and human oversight.

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

## Appendix Contents

## A. Proof of Theorems

### A.1. Proof of Theorem 3.1

*Proof.* We can write

$$
\begin{aligned}
S_{\tau_i^+} &- S_{\tau_i^-} \\
&= \sum_{j=1}^m S_{\tau_i^+,j}/m - \sum_{j=1}^m S_{\tau_i^-,j}/m \\
&= \sum_{j=1}^m (S_{\tau_i^+,j} - S_{\tau_i^-,j})/m \\
&= \sum_{j=1}^m X_j/m,
\end{aligned}
$$

where $X_j = S_{\tau_i^+,j} - S_{\tau_i^-,j}, 1 \le j \le m$. Now we obtain the distribution of $X_j$. We have

$$P(X_j = 2) = P(S_{\tau_i^+,j} = 1, S_{\tau_i^-,j} = -1) = P(S_{\tau_i^+,j} = 1)P(S_{\tau_i^-,j} = -1) = p_+ p_-,$$

$$P(X_j = -2) = P(S_{\tau_i^+,j} = -1, S_{\tau_i^-,j} = 1) = P(S_{\tau_i^+,j} = -1)P(S_{\tau_i^-,j} = 1) = (1-p_+)(1-p_-),$$

$$\text{and } P(X_j = 0) = p_+(1 - p_-) + p_-(1 - p_+),$$

where $p_+ = P(S_{\tau_i^+, j} = 1)$ and $p_- = P(S_{\tau_i^-, j} = -1)$. Therefore mean $\mathbb{E}[X_j] = 2(p_+ + p_- - 1)$, and variance $Var[X_j] \le 4$. Note that $X_j$ are independent. The strong law of large numbers (SLLN) tells us

$$S_{\tau_i^+} - S_{\tau_i^-} \to 2(\mu - 1).$$

So we have

- If $\mu > 1$, then $S_{\tau_i^+} - S_{\tau_i^-}$ is eventually positive almost surely, so $P(S_{\tau_i^+} - S_{\tau_i^-} > 0) \to 1$.

- If $\mu < 1$, then $S_{\tau_i^+} - S_{\tau_i^-}$ is eventually negative a.s., so $P(S_{\tau_i^+} - S_{\tau_i^-} > 0) \to 0$.

- If $\mu = 1$, $P(S_{\tau_i^+} - S_{\tau_i^-} > 0) \to 0.5$ by symmetry of the CLT limit distribution.

Moreover, by Hoeffding's inequality, we have

$$
\begin{aligned}
P(S_{\tau_i^+} > S_{\tau_i^-}) &= P(\sum_{j=1}^{m} X_j > 0) \\
&= P(\sum_{j=1}^{m}(X_j - 2\mu + 2) > -2m(\mu - 1)) \\
&\ge 1 - e^{-m^2(\mu-1)^2/(4m)} \\
&= 1 - e^{-m(\mu-1)^2/4}.
\end{aligned}
$$

$\square$

### A.2. Proof of Theorem 3.2 and Remarks

*Proof.* For each process reward model's evaluative responses which assigns reward $S_{\tau_i, j}$ to step $\tau_i$, the reward for it is $R_{S_{\tau_i, j}} = R_{\tau_i} S_{\tau_i, j}$. So the objective with respect to $S_{\tau_i, j} \sim r_\phi(\cdot \mid \tau_i)$ converges to

$$
\begin{aligned}
&\mathop{\mathbb{E}}_{\substack{q \sim Q \\ \tau \sim \pi_\theta(\cdot|q)}} \mathbb{E}_{S_{\tau_i, j} \sim r_\phi(\cdot|\tau_i)}\big[R_{S_{\tau_i, j}}\big] \\
&= \mathop{\mathbb{E}}_{\substack{q \sim Q \\ \tau \sim \pi_\theta(\cdot|q)}} \mathbb{E}_{S_{\tau_i, j} \sim r_\phi(\cdot|\tau_i)}\big[S_{\tau_i, j}(O_\tau + \mathbb{E}_{S_{\tau_i, l} \sim r_{old}(\cdot|\tau_i)}[S_{\tau_i, l}])\big].
\end{aligned}
$$

Given condition $O_\tau = 1$, then

$$O_\tau + \mathbb{E}_{S_{\tau_i, l} \sim r_{old}(\cdot|\tau_i)}[S_{\tau_i, l}] = 1 + 2P(S_{\tau_i, l} = 1 \mid r_{old}, O_\tau = 1) - 1 = 2P(S_{\tau_i, l} = 1 \mid r_{old}, O_\tau = 1),$$

denoted as $2p_{old}(S_{\tau_i^+, l} = 1)$. $\tau_i^+$ means conditional on $O_\tau = 1$.

Similarly, given condition $O_\tau = -1$,

$$O_\tau + \mathbb{E}_{S_{\tau_i, l} \sim r_{old}(\cdot|\tau_i)}[S_{\tau_i, l}] = -1 - 2P(S_{\tau_i, l} = -1 \mid r_{old}, O_\tau = -1) + 1 = -2P(S_{\tau_i, l} = -1 \mid r_{old}, O_\tau = -1),$$

denoted as $-2p_{old}(S_{\tau_i^-, l} = -1)$. Therefore, the objective is

$$
\begin{aligned}
&\mathop{\mathbb{E}}_{\substack{q \sim Q \\ \tau \sim \pi_\theta(\cdot|q)}} \big[2p_{old}(S_{\tau_i^+, l} = 1)(2P(S_{\tau_i^+, j} = 1) - 1)\mathbb{K}(O_\tau = 1) \\
&\quad - 2p_{old}(S_{\tau_i^-, l} = -1)(2P(-S_{\tau_i^-, j} = -1) + 1)\mathbb{K}(O_\tau = -1)\big] \\
&= 4\mathop{\mathbb{E}}_{\substack{q \sim Q \\ \tau \sim \pi_\theta(\cdot|q)}} \big[p_{old}(S_{\tau_i^+, l} = 1)P(S_{\tau_i^+, j} = 1)\mathbb{K}(O_\tau = 1) \\
&\quad + p_{old}(S_{\tau_i^-, l} = -1)P(S_{\tau_i^-, j} = -1)\mathbb{K}(O_\tau = -1)\big] + C \\
&= 4\mathbb{E}_{q \sim Q}\mathbb{E}_{\tau \sim \pi_\theta(\cdot|q)}\big[p_+ p_{old}(S_{\tau_i^+, l} = 1)\mathbb{K}(O_\tau = 1) + p_- p_{old}(S_{\tau_i^-, l} = -1)\mathbb{K}(O_\tau = -1)\big] + C \\
&= 4\mathbb{E}_{q \sim Q}\big[\langle p_+, f_+\rangle + \langle p_-, f_-\rangle\big] + C,
\end{aligned}
$$

where $f_+ := f_+(\tau) = p_{old}(S_{\tau_i^+,l} = 1)\mathbb{K}(O_\tau = 1) \geq 0$ and $f_- := f_-(\tau) = p_{old}(S_{\tau_i^-,l} = -1)\mathbb{K}(O_\tau = -1) \geq 0$. So if $P(O_\tau = -1 \mid q, \pi_\theta) \to 1$, $\|f_+\|/\|f_-\| \to 0$; if $P(O_\tau = 1 \mid q, \pi_\theta) \to 1$, $\|f_+\|/\|f_-\| \to \infty$.

Note that $C = -2\mathbb{E}_{\substack{q \sim Q \\ \tau \sim \pi_\theta(\cdot|q)}} \left[ p_{old}(S_{\tau_i^+,l} = 1)\mathbb{K}(O_\tau = 1) + p_{old}(S_{\tau_i^-,l} = -1)\mathbb{K}(O_\tau = -1) \right]$, which is irrelevant with $\phi$ we are optimizing this reinforcement learning step.

$\square$

**Remark 1. Different Choices for $\lambda$** When $\lambda = 1$ (our default), $f_+$ and $f_-$ are nonnegative importance-sampling weights for $p_+$ and $p_-$, respectively. As $\lambda$ increases, the policy signal shifts from outcome reward toward process reward, and the reward-model supervision shifts from outcome prediction toward self-consistency. For $\lambda > 0$, $f_+ = 1 + 2\lambda\,p_{old}(S_{\tau_i^+,l} = 1) - \lambda$ and $f_- = 1 + 2\lambda\,p_{old}(S_{\tau_i^-,l} = -1) - \lambda$. Thus, as long as $p_{old}(S_{\tau_i^+,l} = 1) \geq \frac{1}{2}$ and $p_{old}(S_{\tau_i^-,l} = -1) \geq \frac{1}{2} \geq \frac{\lambda-1}{2\lambda}$, the reward system still optimizes the reward model's ability to predict future outcomes. The condition $p_{old}(S_{\tau_i^+,l} = 1)$, $p_{old}(S_{\tau_i^-,l} = -1) \geq \frac{1}{2}$ simply requires the process reward to be better than random guessing, which is mild; actually, $\frac{1}{2} \geq \frac{\lambda-1}{2\lambda}$ holds automatically for $\lambda > 0$. Together, these observations highlight the robustness of our reward system. Also see the ablation results on $\lambda$ (Section 4.9), which illustrate the trade-off between outcome-conditioned and self-consistency supervision.

**Remark 2. Reward Design for Coding Tasks.** The reward design for coding tasks in our experiments also fits within the general RLAnything framework. Recall from Section 4.8 that, for each generated unit test (UT), if it is ground truth (gt), its reward is the number of non-gt code solutions that it causes to fail; otherwise, its reward is the negative number of non-gt code solutions that it incorrectly allows to pass. Since the rewards are ultimately standardized across all generated unit tests for each task, we can formalize the reward as $\hat{p}_c\hat{p}_d - (1 - \hat{p}_c)(1 - \hat{p}_d)$, where $\hat{p}_c = \mathbb{K}\{\text{UT is gt}\}$ and $\hat{p}_d$ is the proportion of non-gt code solutions that the UT detects. This simplifies to $\hat{p}_c + \hat{p}_d - 1$. At the population level, $\mathbb{E}[\hat{p}_c + \hat{p}_d] = p_+ + p_-$, which is exactly the quantity we aim to optimize.

## B. Additional Experimental Results

### B.1. Agentic Coding Applications with Our Optimized Models

We evaluate our optimized coding models under several agentic coding methods, including MPSC (Huang et al., 2024), AlphaCodium (Ridnik et al., 2024), and $S^\star$ (Li et al., 2025a).

In MPSC, we generate 8 samples of code, unit tests, and specifications. A specification is a pair of functions (a pre-condition and a post-condition) that define the valid input space and the expected input-output behavior of a program, serving as a formal description of its intended functionality. We then follow the iterative optimization procedure to compute consistency scores, which are used to identify the best code solution.

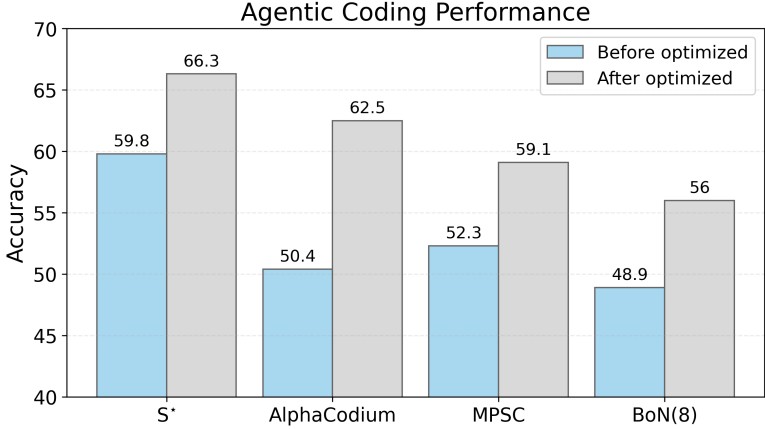

*Figure 7.* Results of agentic coding on LiveBench using models before and after optimization.

In AlphaCodium, we generate 8 code solutions per task using reasoning over public tests, along with 8 corresponding unit tests. Each solution undergoes two refinement iterations based on execution results from the public tests, followed by two additional iterations based on execution results on the generated unit tests. Concretely, each refinement step conditions on the unit tests, the current code, and the execution logs, and decides whether and how to update the solution.

In $S^\star$, we generate 8 code solutions and apply four iterations of self-debugging using public tests to obtain 8 refined versions. Since debugging relies on execution results from ground-truth unit tests, we directly prompt the model to revise the code when a test fails. The final solution is selected using their pairwise comparison method, with generated unit tests as the evaluation signal.

We also consider the most simple test time scaling, method, best of N. Specifically, we independently generate 8 codes and 8 unit tests, and select the code that passes the most generated unit tests as the final solution.

From Figure 7, we find that the optimized models significantly improve agentic coding performance. Before optimization, we use Qwen2.5-7B-Instruct as the policy model and Qwen2.5-14B-Instruct as the reward model (unit-test generator).

*Table 3.* Reward model's step-wise accuracy (%) under different supervision sources across GUI agent and LLM agent settings.

| Supervision Source | GUI Agent | | LLM Agent | |
|---|---|---|---|---|
| | Qwen3-VL-32B-Thinking | OpenCUA-72B | Qwen3-32B | gpt-oss-20b |
| Policy | 86.0 | 83.4 | 47.0 | 40.5 |
| Policy + Reward | 88.7 | 86.2 | 55.2 | 49.8 |
| Policy + Reward + Env | **91.3** +5.3 | **89.9** +6.5 | **59.4**+12.4 | **52.2**+11.7 |

*Table 4.* Reward model's step-wise accuracy (%) under different trajectory sources across GUI agent and LLM agent settings.

| Trajectory Source | GUI Agent | | LLM Agent | |
|---|---|---|---|---|
| | Qwen3-VL-8B-Thinking | OpenCUA-7B | Qwen2.5-7B-Instruct | LLaMA-3.1-8B-Instruct |
| Policy | 86.0 | 84.7 | 47.0 | 56.5 |
| Policy + Reward | 88.7 | 87.6 | 55.2 | 67.1 |
| Policy + Reward + Env | **91.3** +5.3 | **89.6** +4.9 | **59.4**+12.4 | **72.7**+16.2 |

### B.2. Ablation Studies on Using Different Models for Reward Model Evaluation

In this section, we show that using different supervisor models to evaluate the accuracy of assigned step-wise rewards leads to the same conclusion. We provide details of this evaluation approach in Appendix C.3. We conduct ablation studies using OpenCUA-72B in the GUI setting and gpt-oss-20b (OpenAI, 2025) in the LLM-agent setting. In addition, we vary the models used to generate the trajectories that the reward model evaluates: we use OpenCUA-7B for the GUI setting and LLaMA-3.1-8B-Instruct (Dubey et al., 2024) for the LLM-agent setting. The results (Tables 3 and 4) are consistent with our main results in Table 1, validating our approach of using an LLM-as-a-judge for evaluation.

### B.3. Effect of Self-Consistency Reward Signal

We further examine whether the self-consistency component in Eq. 2 is necessary. In addition to the ablation on $\lambda$ in Section 4.9, we conduct an experiment in the ALFWorld setting with $\lambda = 0$, where the reward model is trained using only the outcome signal. We compare it with the full self-consistency setting ($\lambda = 1$), keeping the integrated policy feedback unchanged. After 100 training steps, removing the self-consistency signal leads to a clear drop in step-wise accuracy, from 54.7 to 48.2, while outcome accuracy remains comparable (61.0 vs. 62.7). Moreover, the final policy test accuracy improves from 47.8 under $\lambda = 0$ to 54.1 under $\lambda = 1$. These results suggest that the proposed self-consistency signal is important for learning a better process reward model and can further benefit policy optimization.

*Table 5.* Ablation of the self-consistency reward signal in ALFWorld.

| $\lambda$ | Outcome Acc. | Step Acc. |
|---|---|---|
| 0 | 62.7 | 48.2 |
| 1 | 61.0 | 54.7 |

### B.4. Comparison with Easy-to-Hard Curriculum

We compare our environment adaptation strategy with a simple easy-to-hard curriculum baseline on OSWorld. Specifically, we rank tasks within each category according to the policy model's accuracy and split them into four difficulty groups using accuracy thresholds of 1.0, 0.75, 0.5, and 0.25. The model is then trained for 160 steps following this curriculum. The final accuracy of this baseline is 34.6, which is lower than the 37.5 achieved by our method. This result suggests that our approach provides a more effective adaptation strategy than a straightforward difficulty-based curriculum.

### B.5. Examples of Environment Adaptation

The following GUI-agent example illustrates how the environment model uses summarized error patterns to add tips to the task prompt, making the task easier for the policy. As a result, the policy's accuracy on this task increases from 0 to 0.125. The original task is too difficult for the policy, yielding no successful trajectories and thus no training signal from success cases. After adding tips to simplify the task, the policy achieves occasional successes, providing learning signal and enabling it to gradually solve the task.

---

**GUI Task Adaptation Example 1 (Target: Easier)**

**template:** Using the Pivot Table feature, summarize the total revenue for each promotion type in a new sheet ('Sheet2'), with the promotion names as the column headers.
**old task prompt:** Summarize the total revenue for each promotion type in a new sheet (Sheet2) with the promotion names as the column headers using the Pivot Table feature.
**new task prompt:** Using the Pivot Table feature, summarize the total revenue for each promotion type in a new sheet ('Sheet2'), with the promotion names as column headers. Ensure 'Promotion' is in Column Fields and use 'Insert', 'PivotTable' for correct setup. Verify field names to avoid confusion between 'Date' and 'Promotion'.
**policy accuracy change:** $0 \rightarrow 0.125$ (easier for the policy)

---

The following example shows a case where the GUI task is too easy for the policy model, so the perturbation makes the task harder by switching to a more challenging task template.

---

**GUI Task Adaptation Example 2 (Target: Harder)**

**template candidates:**

- **template1:** Copy the "Revenue" column along with the header to a new sheet named "Sheet2".

- **template2:** Copy the "Revenue" column along with the header to a new sheet named "Sheet2". Then rename this "Revenue" column in "Sheet2" to "Profit".

**old task prompt:** Copy the "Revenue" column along with the header to a new sheet named "Sheet2".
**new task prompt:** Copy the "Revenue" column along with the header to a new sheet named "Sheet2". Then rename this "Revenue" column in "Sheet2" to "Profit".
**policy accuracy change:** $1.0 \rightarrow 0.625$ (harder for the policy)

---

The following two examples adjust task difficulty by varying the number of objects involved.

---

**AlfWorld Task Adaptation Example 1 (Target: Easier)**

**task:** Modify the goal tokens (object type, receptacle type) to make the task **EASIER** while preserving the original goal framework.
**response:** I need to find a pair of object and receptacle types that satisfy the constraints and are more likely to be successfully completed given the current state.
Looking at the "cleanable" list, many objects are cleanable. The "canContain" list shows that many receptacles can hold many objects.
The original goal was to clean PotatoType and place it in MicrowaveType. While valid, this might be less likely to succeed because: 1. PotatoType has only 2 instances (potato1, potato2) 2. MicrowaveType can only hold certain

---

objects (AppleType, BowlType, BreadType, CupType, EggType, MugType, PlateType, PotatoType, TomatoType) 3. The agent is at loc32, which might not be near the microwave

A better choice would be an object with more instances and a receptacle that can hold it. Looking at the "cleanable" list, "CupType" has 3 instances (cup1, cup2, cup3) and can be placed in many receptacles including MicrowaveType. The goal "clean CupType then place into MicrowaveType" would be: - More likely to succeed due to more instances - Valid because CupType is cleanable and MicrowaveType can contain it - Follows the framework of cleaning an object and placing it in a receptacle

**Final Answer**

$\boxed{CupType, MicrowaveType}$

**policy accuracy change:** $0.125 \rightarrow 0.5$

---

**AlfWorld Task Adaptation Example 2 (Target: Harder)**

**task:** Modify the goal tokens (object type, receptacle type) to make the task **HARDER** while preserving the original goal framework.

**response:** Let me analyze the requirements for creating a harder goal.

The original goal is to place a RemoteControlType into an ArmChairType. This is possible because (ArmChairType, RemoteControlType) is in the canContain list.

To make this harder, I need to: 1. Choose an object type with fewer instances 2. Choose a receptacle type that can contain it but has fewer instances 3. Ensure the pair is valid (satisfies canContain)

Looking at the object types: - WatchType has 3 instances (watch1, watch2, watch3) - KeyChainType has 3 instances (keychain1, keychain2, keychain3) - CreditCardType has 1 instance (creditcard1)

Looking at receptacle types: - ArmChairType has 2 instances (armchair1, armchair2) - CoffeeTableType has 2 instances (coffeetable1, coffeetable2) - DiningTableType has 1 instance (diningtable1) - SofaType has 1 instance (sofa1)

The pair (KeyChainType, ArmChairType) is valid (in canContain), and: - KeyChainType has 3 instances - ArmChairType has 2 instances

This is a valid pair, but it's not harder than the original goal.

Let's try (CreditCardType, ArmChairType): - CreditCardType has 1 instance - ArmChairType has 2 instances - (ArmChairType, CreditCardType) is in canContain

This pair is valid and has fewer instances, making it harder.

So the harder goal would be:

**Final Answer**

$\boxed{CreditCardType, ArmChairType}$

**policy accuracy change:** $1.0 \rightarrow 0.25$

---

# C. Experimental Details

## C.1. Models and Settings

For GUI agents applied to OSWorld (Xie et al., 2024), we use Qwen3-VL-8B-Thinking (Bai et al., 2025), UI-TARS1.5-7B (Qin et al., 2025), and OpenCUA-7B (Wang et al., 2025b) in our experiments. For final evaluation, we set the maximum number of steps to 50 and the temperature to 0, and report the average accuracy over 3 independent runs. During RL optimization, we set the maximum number of steps to 30 and use a temperature of 1.0 for the policy model and 0.8 for the process reward model. For the policy model, at each RL step we sample 12 tasks, each with 8 independent rollout trajectories. Context management follows the standard OSWorld pipeline (Xie et al., 2024) by including the three most recent images and summarizing all previous actions as context. When using OpenCUA, we set the CoT level to 2, following its default setting. For the reward model, we use Qwen3-VL-8B-Thinking as the base model and perform 3 evaluations for each policy response. The reward context is constructed by summarizing all previous actions and including the two most recent images, along with the action between those two images which we ask reward model to evaluate. We use Qwen3-4B (Yang et al., 2025) to adapt tasks. We use 12 nodes to conduct the training.

For the LLM agent applied to AlfWorld (Shridhar et al., 2020; Côté et al., 2018), we use Qwen2.5-7B-Instruct as the policy

model and Qwen2.5-14B-Instruct as the reward model (Yang et al., 2024a), and we use Qwen3-4B (Yang et al., 2025) to adapt tasks. For final evaluation, we set the maximum number of steps to 60 and the temperature to 0.8, and report the average accuracy over 3 independent runs. During RL optimization, we set the maximum number of steps to 40 and use a temperature of 0.8 for both the policy model and the process reward model. At each RL step, we sample 16 tasks, each with 8 independent rollouts. We construct the policy-model context by summarizing all previous actions and their corresponding observations, and including the most recent action to be chosen. We use 8 nodes to conduct the training.

For coding LLMs, we use the same model combination as in the LLM-agent setting above. At each RL step, we sample 64 tasks, with 32 independent code-solution generations and 32 independent unit-test generations per task.

During all RL training runs, we use the following standard hyperparameters in the policy objective: clipping threshold $\epsilon = 0.2$ (Schulman et al., 2017), KL-divergence weight $\beta = 0.01$, and a learning rate of $1 \times 10^{-6}$. We use the k3 KL estimator and optimize with AdamW (Loshchilov & Hutter, 2017). We train for 240 steps for the GUI agent, 200 steps for the LLM agent, and 300 steps for the coding LLM to obtain the final models reported in Table 1.

### C.2. Training and Evaluation Datasets

For the GUI agent, the training set excludes the Multiple Apps and Chrome task categories, which are treated as OOD in the final evaluation. As a result, we evaluate on 230 in-domain tasks and 139 OOD tasks. For AlfWorld, the tasks are split into 3.5k training tasks, 140 in-domain evaluation tasks, and 134 OOD evaluation tasks. We use LiveCodeBench-V2 (Jain et al., 2024), CodeContests (Li et al., 2022), and LiveBench (White et al., 2024) for evaluation, and CodeContests (Li et al., 2022) for training. Specifically, for CodeContests, we extract tasks with difficulty level $\leq 2$ and randomly split them into a training set of 4.5k examples and an evaluation set of 200 examples.

### C.3. Evaluation for Reward Models

For OSWorld and AlfWorld settings, we evaluate both the step-wise quality (process accuracy) and the ability to predict a step's influence on the final outcome (outcome accuracy). For outcome accuracy, the ground truth label is simply the verifiable outcome. For process accuracy, the label is provided by a stronger reasoning model than the reward model used.

Specifically, in the OSWorld setting, we use Qwen3-VL-32B-Thinking to provide eight independent evaluations for each policy response using the prompt in Section C.5, and take the majority vote (1 or $-1$) as the ground-truth label. Policy responses are generated by Qwen3-VL-8B-Thinking on the OSWorld multiple apps tasks: for each task, the policy samples 16 independent rollouts, producing 16 trajectories per task. In the AlfWorld setting, we use Qwen3-32B for evaluation and Qwen2.5-7B-Instruct to generate trajectories on the AlfWorld OOD evaluation set (Section C.2), using the same protocol and hyperparameters as in the OSWorld setting.

In the coding setting, we use Qwen2.5-7B-Instruct as the policy model to generate 16 independent code solutions, and Qwen2.5-14B-Instruct as the reward model to generate 32 independent unit tests for each task in the evaluation datasets (LiveCodeBench, CodeContests, or LiveBench). A generated solution is labeled as ground-truth-correct if it passes all dataset-provided unit tests. A generated unit test is correct if it passes on all ground-truth-correct solutions, and is perfect if it is correct and also rejects all non-ground-truth solutions (i.e., causes them to fail). We evaluate the reward model using both the correctness rate and the perfect rate, where the latter corresponds to the detect accuracy reported in Table 1.

### C.4. Policy Prompt Templates

We use these templates for context management in both RL sampling and final evaluation. For OpenCUA and UI-TARS, we follow the standard OSWorld pipeline, which summarizes previous actions while retaining the three most recent images as context.

---

**GUI Agent Prompt Templates (Qwen3-VL-8B-Thinking)**

**Tool-Calling; System Prompt**

```
'''<|im_start|>system
# Tools
You may call one or more functions to assist with the user query.
```

```
You are provided with function signatures within <tools></tools> XML tags:
<tools>
{{tools_def}}
</tools>

For each function call, return a json object with function name and arguments
    within <tool_call></tool_call> XML tags:
<tool_call>
{"name": <function-name>, "arguments": <args-json-object>}
</tool_call>

# Response format
Response format for every step:
1) Action: a short imperative describing what to do in the UI.
2) A single <tool_call>...</tool_call> block containing only the JSON:
    {"name": <function-name>, "arguments": <args-json-object>}.

Rules:
- Output exactly in the order: Action, <tool_call>.
- Be brief: one sentence for Action.
- Do not output anything else outside those parts.
- If finishing, use action=terminate in the tool call.
<|im_end|>
'''
```

**Message Construction**

```
# We construct a multimodal message list as follows:
# 1) A system message containing the tool-calling specification and the tool
    schema.
# 2) For historical context, we keep:
#    - All past actions as a text-only history (Step 1: ..., Step 2: ..., ...).
#    - At most the most recent 3 screenshots (image-only history).
# 3) For each retained past step i:
#    - Append a user message with the screenshot i.
#    - Append an assistant message with the model's response at step i (Action + <
    tool_call>).
# 4) For the current step:
#    - Append a user message containing the current screenshot + the instruction
    prompt
#       (which includes the instruction and the full action history).

# Variables used in the paper template:
# - tools_def: JSON string of tool definitions (i.e., json.dumps(tools_def))
# - step_index: current step id (0-based)
# - screenshots[i]: base64-encoded PNG screenshot at step i (string without the
    data: prefix)
# - responses[i]: assistant response text at step i (Action + <tool_call>)
# - actions: list of action strings taken so far (for the text-only action history)

# - instruction: the current task instruction (string)

messages = [
  {"role": "system", "content": [{"type": "text", "text": system_prompt}]}
]

# Keep at most the last 3 screenshots
start_i = max(0, step_index - 3 + 1)

for i in range(start_i, step_index):
  # Historical screenshot i
  img_url_i = f"data:image/png;base64,{screenshots[i]}"
```

```
  messages.append(
    {"role": "user", "content": [{"type": "image_url", "image_url": {"url":
    img_url_i}}]}
  )
  # Historical assistant response i (Action + <tool_call>)
  messages.append(
    {"role": "assistant", "content": [{"type": "text", "text": responses[i]}]}
  )

# Text-only full action history
previous_actions_str = "None" if len(actions) == 0 else "\n".join(
  [f"Step {k+1}: {a}" for k, a in enumerate(actions)]
)

instruction_prompt = f"""
Please generate the next move according to the UI screenshot, instruction and
    previous actions.

Instruction: {instruction}

Previous actions:
{previous_actions_str}
"""

# Current screenshot + instruction prompt
curr_img_url = f"data:image/png;base64,{screenshots[step_index]}"
messages.append(
  {"role": "user", "content": [
    {"type": "image_url", "image_url": {"url": curr_img_url}},
    {"type": "text", "text": instruction_prompt},
  ]}
)
```

The following is the prompt template for the policy model in the AlfWorld setting, which summarizes all previous actions and the corresponding observations as context.

---

**LLM Agent Prompt Templates (AlfWorld)**

**Guide Prompt)**

```
guide = (
  "You are playing a text game. Your objective is to complete the task as soon as
    possible.\n"
  "Below is your trajectory so far and current candidate actions.\n"
  "You need to think step by step then put the integer (the index of your chosen
    action) in \\boxed{}.\n"
)
```

**Trajectory Rendering; summarized history**

```
# The trajectory is summarized into alternating observation/action lines.
def _render_traj(traj):
  lines = []
  for t in traj:
    if "obs" in t and t["obs"] is not None:
      lines.append(f"observation: {t['obs']}")
    if t.get("act") is not None:
      lines.append(f"you took action: {t['act']}")
  return "\n".join(lines)

trajectory_history = _render_traj(traj)
```

**Full Prompt Template**

```
'''<|im_start|>system
You are a helpful assistant.
<|im_end|>
<|im_start|>user
{{guide}}

{{trajectory_history}}

You need to think step by step then choose one action by number:
{{action_options}}
<|im_end|>
<|im_start|>assistant
'''
```

The following is prompt template used in coding settings.

Coding LLM Prompt Templates

```
'''<|im_start|>system
You are a helpful assistant that helps the user solve programming problems.
<|im_end|>
<|im_start|>user
You need to think first then write a Python script.
You should use input() to read input and print() to produce output in your script.
This is the problem:
{{problem}}

You should put your code in '''python '''.
<|im_end|>
<|im_start|>assistant
'''
```

## C.5. Process Reward Model Prompt Templates

For the GUI-agent setting, we use Qwen3-VL-8B-Thinking as the process reward model to evaluate each policy response. The reward-model context consists of a summary of all previous actions, the two most recent images, and the action between those images that we evaluate. For the AlfWorld setting, we provide the policy prompt and response, and ask the LLM to judge the response. In both settings, the prompts are designed to assess step-wise quality as well as the step's potential consequences for the final outcome. The final output reward can only be 1 or -1. For the coding-LLM setting, we prompt the LLM to generate unit tests. The quality of these unit tests serves as an evaluation signal for certain aspects of the generated code and can be used as a specialized form of process reward.

The following is the prompt template for reward model in OSWorld's setting.

GUI Agent Rewarding Prompt Templates

```
# We build reward_messages as a multimodal user content list.
# The reward prompt includes:
# - A text-only prefix containing "Previous Actions" (older action history).
# - A short window of recent steps: for each step i in the window,
#     (a) the environment screenshot at step i,
#     (b) the agent action taken at step i.
# - The current observation screenshot (the state after the most recent action).
# - A strict evaluation instruction describing the agent objective and the most
    recent response.
```

```
# Variables used in the paper template:
# - step_index: current step id (0-based, the "most recent step" is step_index)
# - actions[i]: action text taken at step i
# - instruction: task instruction / objective string
# - response: the agent's most recent response (reasoning + action/tool call)
# - reward_messages: chat message list for the reward model
# - reward_user_content: multimodal user content list (text/image blocks)

reward_user_content = []

# Keep at most the last 2 steps for reward context (excluding the current
    observation).
rstart_i = max(0, step_index - 2 + 1)

# (1) Previous Actions: all actions before rstart_i
prev_lines = []
for i in range(rstart_i):
  prev_lines.append(f"Step {i+1}: {actions[i]}")
previous_reward_actions_str = "\n".join(prev_lines) if prev_lines else "None"

reward_user_content.append({
  "type": "text",
  "text": f"Previous Actions:\n{previous_reward_actions_str}"
})

# (2) Recent step window: for each step i in [rstart_i, step_index)
for i in range(rstart_i, step_index):
  reward_user_content.append({"type": "text", "text": "Image of environment:\n"})
  reward_user_content.append({"type": "image", "image": "<image>"})
  reward_user_content.append({
    "type": "text",
    "text": f"\nAction of agent:\nStep {i+1}:\n{actions[i]}\n"
  })

# (3) Current observation image (after executing the most recent action)
reward_user_content.append({"type": "text", "text": "Agent's current observation:\
    n"})
reward_user_content.append({"type": "image", "image": "<image>"})

# (4) Evaluation instruction (objective + most recent response)
REWARD_INSTRUCTION_TEMPLATE = r"""
You are a strict evaluator to evaluate the most recent step of the agent in the
    following.
Objective of Agent: {instruction}
Agent's most recent step (reasoning + action): {response}
"""

reward_user_content.append({
  "type": "text",
  "text": "\n" + REWARD_INSTRUCTION_TEMPLATE.format(
    instruction=instruction,
    response=response
  )
})

reward_messages.append({"role": "user", "content": reward_user_content})
```

In our evaluation of the step-wise quality predicted by the reward model, we use the following reward instruction template and ask Qwen3-VL-32B-Thinking to provide labels:

---

**REWARD_INSTRUCTION_TEMPLATE for evaluating reward model's step-wise accuracy (OSWorld)**

```
REWARD_INSTRUCTION_TEMPLATE = r"""
You are a strict evaluator to evaluate the most recent step of the agent in the
    following. Focus on the quality of this step.
Objective of Agent: {instruction}
Agent's most recent step (reasoning + action): {response}
"""
```

The following is the prompt template for reward model in AlfWorld's setting.

**LLM Agent Rewarding Prompt Templates (AlfWorld)**

```
'''<|im_start|>system
You are a helpful assistant.
<|im_end|>
<|im_start|>user
You are a judge for an agent acting in a text-based environment.
Evaluate ONE step using:
 - the agent's prompt (observation + candidate actions),
 - its response (reasoning + chosen index), and
 - the environment's next observation after executing that action.

Scoring (binary):
Score 1 if ALL are true:
 (a) The selected action is appropriate for the current observation and task goal
     (it reasonably explores, progresses or completes the task);
 (b) The reasoning is present, relevant, and not self-contradictory (no
     hallucinated objects/locations);
 (c) The chosen index exists in the candidate list, and the resulting next
     observation is consistent with the described action.
Otherwise score -1. Cases include: no reasoning provided; index out of range;
    clearly irrelevant; undoes progress; self-contradictory/hallucinated reasoning;
     or next observation contradicts the action.

Important: think first then put the final score in \\boxed{}.

Agent's prompt:
{{policy_prompt}}

Agent's response:
{{policy_response}}

Next observation after this action:
{{next_obs}}
<|im_end|>
<|im_start|>assistant
'''
```

In our evaluation of the step-wise quality predicted by the reward model (AlfWorld setting), we use the following reward instruction template and ask Qwen3-32B to provide labels:

**Prompt Template for evaluating reward model's step-wise accuracy (AlfWorld)**

```
'''<|im_start|>system
You are a helpful assistant.
<|im_end|>
```

```
<|im_start|>user
You are a judge for an agent acting in a text-based environment.
Evaluate ONE step using:
 – the agent's prompt (observation + candidate actions),
 – its response (reasoning + chosen index), and
 – the environment's next observation after executing that action.

Scoring (binary):
Score 1 if ALL are true:
 (a) The selected action is appropriate for the current observation;
 (b) The reasoning is present, relevant, and not self-contradictory (no
     hallucinated objects/locations);
 (c) The chosen index exists in the candidate list, and the resulting next
     observation is consistent with the described action.
Otherwise score –1. Cases include: no reasoning provided; index out of range;
    clearly irrelevant; undoes progress; self-contradictory/hallucinated reasoning;
     or next observation contradicts the action.

Important: think first then put the final score in \\boxed{}.

Agent's prompt:
{{policy_prompt}}

Agent's response:
{{policy_response}}

Next observation after this action:
{{next_obs}}
<|im_end|>
<|im_start|>assistant
'''
```

## Coding LLM Reward Prompt Template

```
REWARD_TEST_PROMPT = r"""<|im_start|>system
You are a rigorous unit-test designer for coding problems.
You must produce exactly ONE new test example that is correct and discriminative.
<|im_end|>
<|im_start|>user
You need to provide a new test example. A good test example should be completely
    accurate and conform to the problem's format requirements, while also
    possessing enough discriminative power to distinguish correct code from
    incorrect code.

Before providing a test example, you must think carefully and reason step by step
    to derive an input and output you are very confident are correct. For example,
    start by designing an input you can reliably handle, then compute the output
    step by step. If you're unsure about the output, revise or re-design the input
    to ensure accuracy. Directly providing input/output pairs without this process
    is discouraged, as it often results in low accuracy.

Finally, after completing these previous thinking and derivation steps (you should
     not write the final test example unless you have gone through these steps very
     thoroughly), you MUST put your final test example in the following format:

**Test Input:**
```
<put the EXACT stdin content here>
```
```

```
**Test Output:**
```
<put the EXACT stdout content here>
```

**Explanation:** <brief explanation here>

IMPORTANT:
- Output must contain exactly one **Test Input:** block and one **Test Output:**
    block.
- Use triple backticks exactly as shown.
- The test must be self-contained and match the problem format.

Problem: {{problem}}
<|im_end|>
<|im_start|>assistant
"""
```

### C.6. Error Pattern Summarization and Prompt Templates

We identify where the policy might go wrong by summarizing the thinking portion of the process reward model's outputs. For each task, we obtain a few sentences describing the mistakes the policy makes while solving the task. For GUI agent, we first perform step-wise summarization by aggregating the independent evaluations at steps where at least one evaluation score is $-1$ (indicating a potential mistake), producing a step-level summary. We then perform trajectory-wise summarization: for each trajectory, we use the step-wise summaries as context and ask the model to summarize the mistakes made over the entire trajectory. As a result, for each policy trajectory of each task, we obtain a concise summary of the policy's error patterns. For the LLM-agent setting on AlfWorld, we directly use the full trajectory (the agent's actions and the corresponding observations), and highlight those steps where all evaluation scores are $-1$, as the summarization context. We do not perform the two-stage step-wise followed by trajectory-wise summarization used in the GUI-agent setting because the context here is much shorter and straightforward. We use Qwen3-VL-8B-Thinking for the OSWorld setting and Qwen3-4B for the AlfWorld setting. For coding setting, the diagnostic information consists of the unit tests that the generated code fails.

---

**OSWorld (GUI Agent) Error Summarization Prompt Templates**

**Step-wise Summarization (OSWorld GUI)**

```
# Variables:
# - step_index: int, current step id in the trajectory
# - reward_model_responses: List[str] (or List[dict]/mixed), multiple reward-model
      candidates
# - extracted_reward: List[int], aligned per response (e.g., +1 / -1 / 0)

STEP_ERROR_SUMMARY_PROMPT = (
  "You are analyzing one step in a trajectory for an OSWorld/desktop task.\n\n"
  f"step_index: {step_index}\n\n"
  "You are given:\n"
  "- reward_model_responses (multiple candidates)\n"
  "- extracted_reward aligned with responses (+1/-1/0)\n\n"
  "Task:\n"
  "Write ONE high-density summary (<= 2 sentences) explaining why this step was
    judged negative.\n"
  "Be specific about the failure mode (e.g., wrong assumption, misread UI,
    inconsistent with instruction, hallucinated value, skipped constraint).\n\n"
  "Rules:\n"
  "- Do NOT repeat the prompt verbatim.\n"
  "- Final answer MUST be in \\boxed{...} ONLY.\n\n"
  "reward_model_responses:\n"
  f"{json.dumps(reward_model_responses, ensure_ascii=False)}\n\n"
```

```
    "extracted_reward:\n"
    f"{json.dumps(extracted_reward, ensure_ascii=False)}\n"
)
```

**Trajectory-wise Summarization (OSWorld GUI)**

```
# Variables:
# - step_summaries: List[dict] or List[str], step-level summaries for ONE
    trajectory
#   (typically produced by the step-wise prompt above)

TRAJECTORY_ERROR_SUMMARY_PROMPT = (
  "You are given step-level error summaries for ONE trajectory.\n\n"
  "Task:\n"
  "Produce ONE trajectory-level error summary (<= 2 sentences) capturing the main
    recurring failure modes.\n\n"
  "CRITICAL anti-redundancy rule:\n"
  "- Do NOT repeat the same error across different steps.\n"
  "- If multiple steps share the same failure type, mention it ONCE and, if
    helpful, note it as recurring.\n"
  "- Keep language concise but high information density.\n\n"
  "- Do reasoning first, then put Final Answer in \\boxed{...} ONLY.\n\n"
  "step_error_summaries (JSON):\n"
  f"{json.dumps(step_summaries, ensure_ascii=False)}\n"
)
```

**AlfWorld (LLM Agent) Error Summarization Prompt Template**

```
# Variables:
# - max_steps: int, maximum interaction steps allowed
# - task: str, natural-language task description
# - traj_text: str, full trajectory rendered as observation/action sequence
# - steps_information: str, highlighted steps where all reward-model evaluations
    are -1

ALFWORLD_ERROR_SUMMARY_PROMPT = (
  "<|im_start|>You are a helpful assistant. <|im_end|>\n"
  "<|im_start|>user\n"
  "You are analyzing a failed rollout of a policy in a text-based environment.\n"
  f"The rollout did NOT finish the task within {max_steps} interaction steps.\n"
  f"Task (natural language): {task}\n\n"
  "Full trajectory (observation/action sequence):\n"
  f"{traj_text}\n\n"
  f"Some steps that are marked by reward model to be highly possible incorrect: {
    steps_information}\n\n"
  "In at most TWO sentences, explain the most likely reasons the policy failed to
    finish in time.\n"
  "Be concrete (e.g., wrong exploration, looping, wrong target/location,
    inconsistent reasoning, hallucination, etc.).\n"
  "Put the final <=2 sentence summary in \\boxed{} and output NOTHING else.\n"
  "<|im_end|>\n"
  "<|im_start|>assistant"
)
```

## C.7. Environment Modification and Prompt Templates

To obtain new tasks that better match the policy's current capability while preserving the essence of the original tasks (to prevent the task set from drifting too far from the original distribution), we design the following prompt templates for a reasoning model to adapt tasks based on summarized information about the policy's accuracy and the specific errors it makes

on each task.

For the GUI-agent setting, for each task we provide a set of task templates: the original task is always included, and new but highly related templates are occasionally added. Specifically, we pre-create additional task templates for 47 of the 230 training tasks, resulting in 295 task templates in total (see examples in Appendix C.8). In our ablation studies, all of these tasks are included in the training set. We provide information about where the policy is likely to make mistakes on the original task, and ask the model to select a new task template (if applicable) and write a new task prompt based on that template. When the goal is to make the task easier, the model can add tips to the prompt according to the summarized error patterns, enabling a more proactive adaptation for tasks the policy struggles with. When the goal is to make the task harder, the model can remove such tips and make the instruction more ambiguous. The choice of task template can also depend on the target difficulty and the type of perturbation.

For the LLM-agent setting on AlfWorld, we provide the model with the policy's performance on the task, along with basic environment information (e.g., what objects the environment contains, their properties, and where they are placed) to help the model decide how to modify the task. For example, if an original task in the subcategory "pick and place" is too hard for the policy because it cannot find the target object, the environment model replaces the target with an object that is easier to locate. Conversely, if the task is too easy for the policy, the environment model makes the target object harder to find.

---

**GUI Agent Task-Difficulty Adaptation Prompt Template (OSWorld)**

```
# Variables:
# - goal: str, target difficulty direction, e.g., "easier" or "harder"
# - current_task_json: str, JSON string of the current task object
# - task_template_json: str, JSON string of a mapping evaluator_name -> canonical
    instruction
# - traj_summaries_json: str, JSON string of OPTIONAL historical trajectory-level
    error summaries
#   (each trajectory_summary is already deduplicated across steps)

SYSTEM_PROMPT = """<|im_start|>You are a helpful assistant. <|im_end|>
<|im_start|>user
You will help me adjust the difficulty of an OSWorld/desktop task.

You are given:
(1) current_task (JSON)
(2) task_template: a JSON object mapping evaluator_name -> a canonical instruction
     for that evaluator.
(3) previous_rollout_trajectory_summaries: OPTIONAL historical error analyses from
     earlier rollouts.
    - A task may have multiple trajectories (runs).
    - Each trajectory_summary is already deduplicated across steps (no repeated
    same error across steps).

Goal: make the task {{goal}}.

Rules:
- You MAY switch to a different evaluator from task_template (by changing the key),
     OR keep the same evaluator.
- You MAY rewrite the instruction to increase/decrease hint strength (add hints to
     make easier, remove hints to make harder).
- The instruction can NOT be too long.
- You MUST NOT change the essential task meaning compared to the chosen evaluator'
    s template. Do NOT invent a new task.
- Do NOT invent new evaluator names. The output key must be one of the keys in
    task_template.
- You SHOULD use previous_rollout_trajectory_summaries to guide how you adjust
    difficulty:
    - If goal is EASIER: add minimal, targeted clarifying hints addressing recurring
     failure modes.
    - If goal is HARDER: remove such hints, but still keep the same essential task
    and stay within the chosen evaluator template.
```

```
- Output MUST be valid JSON ONLY (no markdown, no extra text).
- Output format MUST be the new current_task JSON object with EXACTLY ONE key:
  {"evaluatorX": "your rewritten instruction"}
- If you accidentally output other text, ensure the FINAL output segment is the
   JSON object.

current_task:
{{current_task_json}}

task_template:
{{task_template_json}}

previous_rollout_trajectory_summaries:
{{traj_summaries_json}}
<|im_end|>
<|im_start|>assistant
"""
```

## AlfWorld (LLM Agent) Task-Difficulty Adaptation Prompt Template

**Environment Summary from INIT**

```
# summarize_init_english(problem_text) returns:
# (1) summary_text: a human-readable English summary of INIT facts, including:
#      - object/receptacle types -> concrete instances
#      - locations
#      - canContain(type -> type) constraints
#      - other instantiated predicates grouped by predicate name
# (2) S: a structured dict of parsed facts for downstream constraints, including:
#      - obj2otype, rec2rtype
#      - otype2objs, rtype2recs
#      - canContain
#      - caps (capability sets like pickupable/toggleable/cleanable/heatable/
#    coolable/sliceable)

summary_text, S = summarize_init_english(problem_text)
```

**Prompt Construction (environment info + failure summaries + goal editing instruction)**

```
# Variables:
# - problem_text: str, the raw problem specification containing INIT facts
# - item["failed_rollout_summaries"]: List[dict], each has fields like rollout_idx
#     and summary
# - goal: str, "harder" or "easier"
# - acc_before: float, previous rollout accuracy (prev_acc)
# - task: str, task subtype name, e.g., "pick_and_place_simple"
# - goal_obj_types, goal_rec_types: List[str], original goal tokens/types
# - S: structured environment facts returned by summarize_init_english(...)
# - goal_brief_and_instruction(...): returns a compact, task-specific editing
#    rubric

summary_text, S = summarize_init_english(problem_text)

prompt_text = (
  "<|im_start|>You are a helpful assistant. <|im_end|>\n"
  "<|im_start|>user\n"
  "Review the following details about an interactive environment. A related task
    will follow.\n"
  + summary_text
)
```

```
prompt_text += "\n\n---\n"

fails = item.get("failed_rollout_summaries", [])
if fails:
  prompt_text += "### Failure summaries from recent rollouts (failed rollouts only)
    \n"
  for rec in sorted(fails, key=lambda x: int(x.get("rollout_idx", 0))):
    rj = rec.get("rollout_idx", 0)
    ss = str(rec.get("summary", "")).strip()
    prompt_text += f"- Rollout {rj}: {ss}\n"
  prompt_text += "\n"

prompt_text += f"### Your job is to propose a new goal that makes the task **{goal.
    upper()}**.\n"
prompt_text += f"- The parent rollout accuracy (prev_acc) is {acc_before}.\n"
prompt_text += "- The new goal must be different from the original and follow the
    instructions.\n"
prompt_text += "- The overall framework of the goal cannot be changed; you may
    only modify two tokens within this framework.\n"
prompt_text += "Represent the new goal by outputting two tokens, placed inside \\
    boxed{} and separated by a comma, e.g., \\boxed{TOKEN_A,TOKEN_B}.\n"
prompt_text += "You need to think step by step then provide final result in \\
    boxed{}.\n"

prompt_text += "\n" + goal_brief_and_instruction(
  task, goal_obj_types, goal_rec_types, S, direction=goal, prev_acc=acc_before
)

prompt_text += "\n<|im_end|>\n<|im_start|>assistant"
```

**Task-Specific Editing Rubric (`goal_brief_and_instruction`)**

```
def goal_brief_and_instruction(task, goal_obj_types, goal_rec_types, S,
                               direction: Optional[str] = None,
                               prev_acc: Optional[float] = None):
    lines = []
    if direction in ("harder", "easier"):
        lines.append(f"### Difficulty goal: **{direction.upper()}** (prev_acc={
    prev_acc})")
        if direction == "harder":
            lines.append("- Prefer types with *fewer* available instances (rarer)
    while keeping constraints satisfied.")
            lines.append("- Prefer combinations likely requiring more search/steps,
     but still solvable in this environment.")
        else:
            lines.append("- Prefer types with *more* available instances (more
    common) while keeping constraints satisfied.")
            lines.append("- Prefer combinations likely easier to find/complete,
    but still valid.")

    if task == "pick_and_place_simple":
        g = (goal_obj_types[0] if goal_obj_types else "<?>",
             goal_rec_types[0] if goal_rec_types else "<?>")
        lines.append("**Overall Framework**: place an object type into/on a
    receptacle type.")
        lines.append(f"**Original goal**: place an object of type **{g[0]}** into/
    on a receptacle of type **{g[1]}**.")
        lines.append(f"The final output example is \\boxed{{{g[0]}, {g[1]}}}")
        lines.append("**Design instruction**: Output exactly **two tokens** - `<
    OBJ_TYPE> <REC_TYPE>`.")
        lines.append("- Constraints: pair must satisfy `canContain(REC_TYPE,
    OBJ_TYPE)`.")
```

```python
    elif task == "look_at_obj_in_light":
        g = (goal_obj_types[0] if goal_obj_types else "<?>",
             goal_obj_types[1] if len(goal_obj_types) > 1 else "<?>")
        lines.append("**Overall Framework**: a light object type is present at the
 agent's location; the agent **holds** an object type.")
        lines.append(f"**Original goal (Example)**: a **toggleable and toggled**
light object of type **{g[0]}** is present; agent **holds** type **{g[1]}**.")
        lines.append(f"The final output example is \\boxed{{{{g[0]}, {g[1]}}}}")
        lines.append("**Design instruction**: Output exactly **two tokens** - '<
LIGHT_OBJ_TYPE> <HOLD_OBJ_TYPE>'.")
        lines.append("- Constraints: LIGHT must have 'toggleable'; HOLD should
have a 'pickupable' instance in INIT.")

    elif task == "pick_clean_then_place_in_recep":
        g = (goal_obj_types[0] if goal_obj_types else "<?>",
             goal_rec_types[0] if goal_rec_types else "<?>")
        lines.append("**Overall Framework**: **clean** an object type and place it
 into/on a receptacle type.")
        lines.append(f"**Original goal**: clean type **{g[0]}** then place into/on
 type **{g[1]}**.")
        lines.append(f"The final output example is \\boxed{{{{g[0]}, {g[1]}}}}")
        lines.append("- Constraints: OBJ must be cleanable; canContain(REC,OBJ).")

    elif task == "pick_heat_then_place_in_recep":
        g = (goal_obj_types[0] if goal_obj_types else "<?>",
             goal_rec_types[0] if goal_rec_types else "<?>")
        lines.append("**Overall Framework**: **heat** an object type and place it
into/on a receptacle type.")
        lines.append(f"**Original goal**: heat type **{g[0]}** then place into/on
type **{g[1]}**.")
        lines.append(f"The final output example is \\boxed{{{{g[0]}, {g[1]}}}}")
        lines.append("- Constraints: OBJ must be heatable; canContain(REC,OBJ).")

    elif task == "pick_cool_then_place_in_recep":
        g = (goal_obj_types[0] if goal_obj_types else "<?>",
             goal_rec_types[0] if goal_rec_types else "<?>")
        lines.append("**Overall Framework**: **cool** an object type and place it
into/on a receptacle type.")
        lines.append(f"**Original goal**: cool type **{g[0]}** then place into/on
type **{g[1]}**.")
        lines.append(f"The final output example is \\boxed{{{{g[0]}, {g[1]}}}}")
        lines.append("- Constraints: OBJ must be coolable; canContain(REC,OBJ).")

    elif task == "pick_two_obj_and_place":
        g = (goal_obj_types[0] if goal_obj_types else "<?>",
             goal_rec_types[0] if goal_rec_types else "<?>")
        lines.append("**Overall Framework**: place **two distinct objects** (same
type) into/on a receptacle type.")
        lines.append(f"**Original goal**: place two objects of type **{g[0]}**
into/on type **{g[1]}**.")
        lines.append(f"The final output example is \\boxed{{{{g[0]}, {g[1]}}}}")
        lines.append("- Constraints: >=2 instances of OBJ_TYPE; canContain(REC,OBJ)
.")

    else:
        lines.append("**Original goal**: (unknown task type).")

    return "\n".join(lines)
```

## C.8. Examples of Task Templates in GUI data

As we discussed in Appendix C.7, we add new task templates to some tasks in the GUI training data. We provide examples as follows. Each task template is coupled with an evaluator and its corresponding verifiable outcome file.

---

**AlfWorld (LLM Agent) Task-Difficulty Adaptation Prompt Template**

```
TASK_TEMPLATE_EXAMPLES = r"""
Example 1:
"task_template": {
  "evaluator1": "Work out the monthly total sales in a new row called 'Total', and
     then create a line chart to show the results (with Months on the x-axis).",
  "evaluator2": "Work out the monthly total sales in a new row called 'Total'.",
  "evaluator3": "Work out January's total sales in a new row called 'Total'.",
  "evaluator4": "Work out the monthly total sales in a new row called 'Total', and
     then create a line chart to show the results (with Months on the x-axis, for
    January, February, and March only)."
}

Example 2:
"task_template": {
  "evaluator1": "Fill all blank cells in B1:E30 with the value from the cell
    directly above. Finish the task and do not modify irrelevant regions, even if
    they are blank.",
  "evaluator2": "Fill all blank cells in B1:B30 with the value from the cell
    directly above. Finish the task and do not modify irrelevant regions, even if
    they are blank.",
  "evaluator3": "Fill all blank cells in E1:E30 with the value from the cell
    directly above. Finish the task and do not modify irrelevant regions, even if
    they are blank.",
  "evaluator4": "Fill all blank cells in E1:E24 with the value from the cell
    directly above. Finish the task and do not modify irrelevant regions, even if
    they are blank."
}

Example 3:
"task_template": {
  "evaluator1": "I have compute the acceleration in row 2 and I want you to fill
    out other rows for column B and D. Next concatenate the values from columns A
    to D, including their headers (the pattern is \"Header: cell value, ..., Header
    : cell value\"), into a new column named \"Combined Data\" for all rows. In the
     new column, only keep 2 decimal digits.",
  "evaluator2": "I have computed the acceleration in row 2, and I want you to fill
     in the remaining rows for columns B and D."
}

Example 4:
"task_template": {
  "evaluator1": "Set the background color to yellow for any slide that contains
    one or more images of real people, and set the title of slide 2 as \"Let's
    start\".",
  "evaluator2": "Set the background color to yellow for any slide that contains
    one or more images of real people",
  "evaluator3": "Set the background color to yellow for slide 2"
}

Example 5:
"task_template": {
  "evaluator1": "I would like to make the first three words of the sentence left-
    aligned and the rest right-aligned. I basically want to have some empty space
    in the middle to add some photos. Assume that every sentence will have at least
     three words. Could you help me on alignment for me using tabstops?",
```

```
  "evaluator2": "I would like to make the first word of the sentence left-aligned
    and the rest right-aligned. I basically want to have some empty space in the
    middle to add some photos. Assume that every sentence will have at least three
    words. Could you help me on alignment for me using tabstops?"
}
"""
```

## C.9. Task Specific Algorithm

---

**Algorithm 2** Task Specific Algorithm Pipeline

---

1: **Given:** environment task set $Q$; policy $\pi_\theta$; reward model $r_\phi$; thresholds $\alpha_{\text{high}}, \alpha_{\text{low}}$.
2: $\mathcal{T} = \emptyset$ is the temporary task set.
3: **for** $k = 1, \ldots, K$ **do**
4:     **Sampling:**
5:     Sample sub set of tasks $\mathcal{S} \subset Q/\mathcal{T}, \mathcal{S}' = \mathcal{S}$.
6:     **if** $k \neq 1$:
7:         $\mathcal{S}' = \mathcal{S} \cup \mathcal{T}$.
8:     **if** task type is OSWorld or AlfWorld.
9:         For each $q \in \mathcal{S}'$, $\pi_\theta$ samples trajectories $\mathbb{T}_q$, each $\tau \in \mathbb{T}_q$, $\tau = (\tau_1, \ldots, \tau_{T_\tau})$. Outcome $O_\tau \in \{-1, 1\}$.
10:         For each $\tau \in \mathbb{T}_q$, reward model samples reasoning $r_{\tau_i,j}$ and final score $S_{\tau_i,j} \in \{-1, 1\}$ for $1 \leq j \leq m$.
11:         Compute step-wise quality for each step $\tau_i$, $R_{\tau_i} = O_\tau + \lambda \sum_{j=1}^m S_{\tau_i,j}/m$    for $\lambda > 0, i = 1, \ldots, T_\tau$.
12:         $R_{\tau_i,j} = R_{\tau_i} \cdot S_{\tau_i,j}$.
13:     **else if** task type is coding.
14:         For each $q \in \mathcal{S}'$, $\pi_\theta$ samples trajectories $\mathbb{T}_q$. $U_q$ is gt UTs set. $O_\tau \in \{-1, 1\}$ is execution result of $\tau$ on $U_q$.
15:         For each $\tau \in \mathbb{T}_q$, reward model samples reasoning $r_{q,j}$ and unit test $u_{q,j}$, $1 \leq j \leq m$.
16:         $O_{q,j} = 1$ if $u_{q,j}$ passes all $\{\tau \in \mathbb{T}_q \mid O_\tau = 1\}$, else -1.
17:         Compute reward for policy, $R_\tau$ = accuracy of code $\tau$ on $U_q \cup \{u_{q,j}\}_{j=1}^m$.
18:         $R_{\tau,j} = 1 - p$ if $O_{q,j} = 1$, $R_{\tau,j} = -p$ if $O_{q,j} = -1$. $p$ is pass rate of $u_{q,j}$ on $\{\tau \in \mathbb{T}_q \mid O_\tau = -1\}$.
19:     **Accept New tasks:**
20:     $\mathcal{N} = \emptyset$ # collect newly accepted tasks.
21:     **for** each $(q, q') \in \mathcal{P}$: # all tasks' accuracy in $\mathcal{P}$ has been updated in the above step
22:         **if** $\text{acc}(q) > \alpha_{\text{high}}$ and $\alpha_{\text{low}} < \text{acc}(q') < \text{acc}(q)$: # $\text{acc}(q') > \alpha_{\text{low}} > 0$, safe
23:             replace $q \leftarrow q'$ in $Q$, and $\mathcal{N} = \mathcal{N} \cup \{q'\}$. # The 'harder' goal has been achieved, accept the new task
24:         **else if** $\text{acc}(q) < \alpha_{\text{low}}$ and $\text{acc}(q) < \text{acc}(q') < \alpha_{\text{high}}$: # $\text{acc}(q') > \text{acc}(q) > 0$, safe
25:             replace $q \leftarrow q'$ in $Q$, and $\mathcal{N} = \mathcal{N} \cup \{q'\}$. # The 'easier' goal has been achieved, accept the new task
26:     $\mathcal{P} = \emptyset, \mathcal{T} = \emptyset$. # ready to update them
27:     **Adapt environment task:**
28:     **for** each $q \in \mathcal{S}$:
29:         $s \leftarrow$ summarize step-wise errors $(\{(\tau_i, r_{\tau_i,j}) \mid 1 \leq i \leq T_\tau, \tau \in \mathbb{T}_q\})$ for $q$.
30:         **if** $\text{acc}(q) > \alpha_{\text{high}}$: # set goal 'harder'
31:             Propose harder task: $q' \leftarrow \text{harder}(q; s)$.
32:             $\mathcal{T} = \mathcal{T} \cup \{q'\}$ and $\mathcal{P} = \mathcal{P} \cup \{(q, q')\}$.
33:         **else if** $\text{acc}(q) < \alpha_{\text{low}}$: # set goal 'easier'
34:             Propose easier task: $q' \leftarrow \text{easier}(q; s)$.
35:             $\mathcal{T} = \mathcal{T} \cup \{q'\}$ and $\mathcal{P} = \mathcal{P} \cup \{(q, q')\}$.
36:     **Update policy $\pi_\theta$:**
37:     **for** $q \in \mathcal{S} \cup \mathcal{N}$:
38:         Compute advantages $A_{\tau_i}^{\pi_\theta}$ by standardize $R_{\tau_i}$ across $\tau \in \mathbb{T}_q$ at same $i$.
39:     Train $\pi_\theta$ with $\{(\tau_i, A_{\tau_i}^{\pi_\theta}) \mid q \in \mathcal{S} \cup \mathcal{N}, \tau \in \mathbb{T}_q, 1 \leq i \leq T_\tau\}$. # each pair is (response, advantage)
40:     **Update reward model $r_\phi$:**
41:     **for** $q \in \mathcal{S} \cup \mathcal{N}$ and $\alpha_{\text{low}} \leq \text{acc} \leq \alpha_{\text{high}}$:
42:         Compute advantages $A_{\tau_i,j}^{r_\phi}$ by standardize $R_{\tau_i,j}$ across $j$, for each $\tau_i$.
43:     Train $r_\phi$ with $\{(r_{\tau_i,j}, A_{\tau_i,j}^{r_\phi}) \mid q \in \mathcal{S} \cup \mathcal{N}, \tau \in \mathbb{T}_q, 1 \leq i \leq T_\tau, 1 \leq j \leq m\}$. # each pair is (response, advantage)
44: **end for**

---

