# OpenReview forum: "RLAnything: Forge Environment, Policy, and Reward Model in Completely Dynamic RL System"
_ICML.cc/2026/Conference — ICML 2026 regular_

### Official Review · Reviewer_FZiR · 2026-03-11

**Soundness:** 3
**Presentation:** 3
**Significance:** 3
**Originality:** 4
**Overall Recommendation:** 5
**Confidence:** 4

**Summary:**

This paper proposes RLAnything, a unified reinforcement learning framework capable of the joint and collaborative optimization of the policy, reward model, and environment. The authors optimize the reward model by leveraging trajectory quality evaluation and step-wise response consistency. For environment optimization, a critic-based approach is employed to integrate steps and adapt task difficulty to meet optimization requirements. Experimental results demonstrate that each component contributes to the final performance of the policy, with evaluations conducted across multiple tasks including GUI and code.Strengths:Proposes a collaborative optimization framework that pushes the upper bound of RL performance.The methods and descriptions provided are mostly actionable and practical for implementation.The authors conducted thorough comparative and ablation studies, clearly illustrating the roles and relationships between various components.Weaknesses:The description regarding the training of the reward model is somewhat vague. The main text provides the reward calculation method but lacks details on the specific optimization algorithms or training procedures.AlfWorld is a relatively dated benchmark. There is a risk of data contamination in modern policy models. Testing on more recent datasets would yield more convincing results.

**Compliance With Llm Reviewing Policy:**

Affirmed.

**Final Justification:**

This paper is a solid one which address a co-optimizing problem for agentic RL. I will maintain my positive evaluation.

**Key Questions For Authors:**

1. Regarding Figure 5: Why does integrated feedback (combining step-wise and outcome signals) appear to underperform compared to using process supervision alone?

2. Given that AlfWorld is an older benchmark, what was the rationale behind choosing it for evaluation?

3. Is the reward model trained online alongside the policy during the RL process? Could you specify the exact optimization method used for its updates?

4. How do you ensure that the new tasks generated via the critic are actually solvable for the agent?

**Limitations:**

The author has discussed the potential negative societal impact of their work. Limitations are presented in the weakness above.

**Strengths And Weaknesses:**

Strengths:
1. Proposes a collaborative optimization framework that pushes the upper bound of RL performance.
2. The methods and descriptions provided are mostly actionable and practical for implementation.
3. The authors conducted thorough comparative and ablation studies, clearly illustrating the roles and relationships between various components.

Weaknesses:
1. The description regarding the training of the reward model is somewhat vague. The main text provides the reward calculation method but lacks details on the specific optimization algorithms or training procedures.
2. AlfWorld is a relatively dated benchmark. There is a risk of data contamination in modern policy models. Testing on more recent datasets would yield more convincing results.
3. Environment adaptation is relatively far-fetched. For the agent, the dynamic changes in its environment are a significant factor in increasing the difficulty of the task, not just the difficulty of the query.

---

> ### Author Rebuttal · Authors · 2026-03-31
>
> Thank you very much for your appreciation of this work!
>
> 1. Regarding Figure 5: Why does integrated feedback appear to underperform compared to using process supervision alone?
>
> In the process-reward-alone curve, the policy is trained using the optimized reward model rather than the reward model before RLAnything training. The reward model used in integrated feedback is the one before training. We use this experiment to demonstrate the superiority of the optimized PRM and to highlight the potential for automatic self-evolving optimization without relying on outcome rewards that depend on human labeling.
>
> 2. AlfWorld is a relatively dated benchmark. What was the rationale behind choosing it for evaluation?
>
> We also evaluated our method on OSWorld, an advanced computer-use benchmark with multi-turn multimodal interactions, and LiveCodeBench, a coding benchmark with single-turn text-only tasks. We chose AlfWorld as a multi-turn text-only setting for LLMs to make the evaluation more comprehensive. These results demonstrate the generalizability of our algorithm across diverse datasets and interaction settings.
>
>
> 3. The description regarding the training of the reward model is somewhat vague. Is the reward model trained online alongside the policy during the RL process? Could you specify the exact optimization method used for its updates?
>
> Sorry for the confusion. The detailed algorithm, including the training of the reward model, is provided in Appendix C.9. The reward model is trained online together with the policy model. We will highlight this more clearly in the main method section to avoid any confusion.
>
> 4. How do you ensure that the new tasks generated via the critic are actually solvable for the agent?
>
> In our environment adaptation process, a generated task q' is accepted if and only if the accuracy of the new task, acc(q'), satisfies the conditions specified in Section 3.4. Importantly, these conditions guarantee that acc(q'), which helps ensure that the accepted task remains solvable for the agent.
>
>
> 5. The dynamic changes in its environment are a significant factor in increasing the difficulty of the task, not just the difficulty of the query.
>
> In our OSWorld setting, the validator files may be modified when adapting to new tasks. This changes the environment, as the file policy the agent needs to focus on is no longer the same, making the new environment more than just a simple change in the query. In future work, we will explore additional engineering methods for automatic environment adaptation.
>
> Please feel free to raise any further questions, and we will do our best to address them.

---

> > ### Author Rebuttal · Reviewer_FZiR · 2026-04-01
> >
> > Most of the answers addressed my concerns. I still have some questions regarding the choice of ALFWorld, as there are many other text-only benchmarks (e.g., WebShop, tau-bench) that are more recent and arguably closer to real-world agent settings.
> > I would appreciate further clarification on why ALFWorld was selected as the primary testbed, and whether the proposed approach is expected to generalize to these more modern environments.

---

> > > ### Author Response · Authors · 2026-04-02
> > >
> > > Thank you very much for your quick reply!
> > >
> > > **Why ALFWorld.** This paper focuses on long-horizon agentic tasks (Section 1, lines 44 left page and 71 right page). WebShop typically involves only 5–10 steps, and tau-bench around 10–30 steps. In contrast, ALFWorld often requires 40–60 steps for reasonable performance (Figure 5(b)), as the policy must sufficiently explore the environment. This makes ALFWorld a more suitable testbed for studying the core challenge we address — credit assignment and curriculum design in long-horizon RL. Additionally, tau-bench's official setup relies on gpt/gemini API calls for non-solo mode, which is not ideal for large-scale RL training under limited API resources.
> > >
> > > **Generalizability by design.** We want to emphasize that RLAnything is not tailored to any specific environment — its three core components are environment-agnostic:
> > >
> > > (1) Process Reward Model: The PRM performs credit assignment by leveraging next-state information to evaluate intermediate steps. This mechanism applies to any sequential decision-making setting regardless of the observation modality (text, GUI, etc.) or step count — whether it is a 5-step WebShop trajectory or a 50-step OSWorld episode.
> > >
> > > (2) Environment Model (Automatic Curriculum): The environment model adjusts task difficulty based on feedback from the reward signal and policy performance. This only requires a parameterized environment that can be prompted or configured to vary difficulty — a property shared by WebShop (e.g., product catalog complexity), tau-bench (e.g., user intent complexity), and virtually any interactive benchmark.
> > >
> > > (3) Policy Optimization: The policy training loop is standard RL and is entirely decoupled from environment specifics.
> > >
> > > To demonstrate this generality beyond text-only settings, we evaluated on OSWorld — a modern, multimodal computer-use benchmark with 30–50 interaction steps — showing that the same framework transfers to real-world application settings without architectural modification.
> > >
> > > We appreciate the reviewer's thoughtful suggestion. We believe the environment-agnostic design of RLAnything, together with the validated generalization on OSWorld, provides strong evidence that the framework naturally extends to WebShop, tau-bench, and other emerging agent benchmarks.
> > >
> > > Please feel free to raise any further questions, and we will do our best to address them.

---

### Official Review · Reviewer_jYEd · 2026-03-12

**Soundness:** 4
**Presentation:** 4
**Significance:** 3
**Originality:** 4
**Overall Recommendation:** 5
**Confidence:** 2

**Summary:**

This paper proposes RLAnything, an RL framework that jointly optimizes policy, reward model, and the environment in a closed loop. The policy is trained using outcome reward and step-wise signals from the reward model, while the reward model is updated with consistency-based objective, and the environment is adapted to be easier or harder using critic feedback. The method is evaluated on various benchmarks, including computer use, coding, which shows improvement to baselines for both in-domain and ood tasks.

**Compliance With Llm Reviewing Policy:**

Affirmed.

**Final Justification:**

The authors have fully addressed my concerns, especially the tight coupling of all components, with experiments.

**Key Questions For Authors:**

1. Can the authors compare critic-guided environment adaptation against simpler baselines, such as 1) adaptation driven only by policy accuracy and 2) rewriting tasks for simpler and harder variants in advance?

2. In table 1 for coding tasks, there are big improvements in terms of unit test accuracy. How are these accuracies measured? How do you make sure that the tests it generate is of high quality in the absence of ground truth solution?

**Limitations:**

Authors have adequately discussed potential negatives societal impact of the work.

**Strengths And Weaknesses:**

Strengths:
* The paper is well written and easy to follow. The high-level pipeline is explained very well by the figures.
* The fully dynamic RL system that jointly optimizes everything (environment, policy, and reward model) is, to the best of my knowledge, novel and important.
* Experiments span several fairly different settings which all show strong performance gains.

Weaknesses:
* The environment adaption should be measured against simple alternatives like easy-to-hard curriculum, so that the source of gains can be clear.
* Closed-loop coupling of all the components may make training much more challenging.

---

> ### Author Rebuttal · Authors · 2026-03-31
>
> Thank you very much for your appreciation of this work!
>
> 1. Can the authors compare critic-guided environment adaptation against simpler baselines, such as adaptation driven only by policy accuracy and rewriting tasks for simpler and harder variants in advance?
>
> We conduct this ablation study to test whether the critic feedback from the reward model is really helpful. Specifically, we only use the accuracy information from the policy model to ask the environment to adapt in the AlfWorld and OSWorld settings. We train for 100 RL steps for AlfWorld and 160 steps for OSWorld. In the OSWorld setting, we find that the accuracy of the policy after optimization drops from 37.5 to 35.3 when not using critic feedback from the reward model. In the AlfWorld setting, we find that the accuracy of the policy after optimization drops from 54.1 to 53.0 when not using critic feedback from the reward model. We then conduct more detailed explorations and find that the number of accepted new tasks drops from 227 to 182 in OSWorld setting, which demonstrates that the critic feedback allows the environment to conduct effective adaptation more easily, since the adaptation becomes more directive and informative.
>
> 2. Measure environment adaption against simple alternatives like easy-to-hard curriculum.
>
> We first rank the tasks in each category of OSWorld based on the accuracy of the policy model, and divide them into four groups using accuracy thresholds of 1.0, 0.75, 0.5, and 0.25. We then train for 160 steps and find that the final average accuracy is 34.6, which is substantially lower than the 37.5 achieved by our environment adaptation approach. This result demonstrates the superiority of our method.
>
> 3. In table 1 for coding tasks, there are big improvements in terms of unit test accuracy. How are these accuracies measured? How do you make sure that the tests it generate is of high quality in the absence of ground truth solution?
>
> For the evaluation of unit test accuracy, we first obtain the ground-truth code for each task. If the ground-truth code is not provided in the original dataset, we ask Qwen3-32B to generate it 16 times. If no generated code for this task passes all the unit tests provided in the original dataset, we do not include this task in the unit test evaluation. After we obtain the ground-truth code, we identify a generated unit test as correct if and only if it passes on the ground-truth code for this task. We find that, before optimization, the unit test generator tends to generate overly difficult unit tests that it itself cannot guarantee to be correct. After optimization, the model first focuses on the correctness of the unit test and then thinks about how to make the unit test harder. We will add these detailed approaches in the appendix.
>
> 4. Closed-loop coupling of all the components may make training much more challenging.
>
> We further scale the training of our GUI model to 400 steps and record the validation accuracy throughout training, as reported in Table 3. The results show that our optimization method remains both scalable and stable. Therefore, the closed-loop coupling does not compromise the stability or scalability of the training process.
>
> **Table 3**
> | Step | Training Acc |
> |------|--------------|
> | 0    | 27.7         |
> | 80   | 34.9         |
> | 160  | 37.6         |
> | 240  | 39.4         |
> | 320  | 41.5         |
> | 400  | 41.4         |
>
>
> Please feel free to raise any further questions, and we will do our best to address them.

---

> > ### Author Rebuttal · Reviewer_jYEd · 2026-04-01
> >
> > I appreciate the authors’ response. My concerns have been completely resolved, and I have raised my score from 4 to 5.

---

> > > ### Author Response · Authors · 2026-04-02
> > >
> > > Dear Reviewer jYEd,
> > >
> > > Thank you very much for raising score! We sincerely appreciate your time, effort, and support of our work!
> > >
> > > Best regards,
> > >
> > > Authors of RLAnything

---

### Official Review · Reviewer_PskM · 2026-03-13

**Soundness:** 3
**Presentation:** 3
**Significance:** 3
**Originality:** 3
**Overall Recommendation:** 4
**Confidence:** 3

**Summary:**

RLAnything is a reinforcement learning (RL) framework that jointly and dynamically optimizes the policy, the reward model, and the environment within a closed-loop system. For policy optimization, it employs integrated feedback that combines verifiable outcome rewards with step-wise signals from the reward model. Reward model training utilizes consistency feedback, aligning step-level evaluations with final outcomes and self-consistency to provide reliable supervision. Finally, the environment leverages critic feedback from both the policy and reward model to automatically adjust task difficulty, ensuring the agent consistently learns from an optimal curriculum of experience.

**Compliance With Llm Reviewing Policy:**

Affirmed.

**Key Questions For Authors:**

Is the specific reward design in Equation 2 strictly necessary for training the reward function? Specifically, how would the performance differ if final outcome correctness were used as the primary training label instead of the proposed consistency signals?

The joint optimization of the policy and reward function resembles Actor-Critic frameworks, such as PPO. Have the authors performed a direct comparison with a PPO baseline, or could they more explicitly discuss the structural differences between these approaches?

**Limitations:**

Yes

**Strengths And Weaknesses:**

1. The reward design of the policy optimization and reward function training make sense to me and the idea seems noval.
2. The authors demonstrate the framework's robustness across three distinct and challenging domains: multimodal GUI agents, text-based interactive games, and coding LLMs.
3. Figure 1 provide a high-level conceptual map and a concrete implementation path, making the complex closed-loop system easier to digest.

---

> ### Author Rebuttal · Authors · 2026-03-31
>
> Thank you very much for your appreciation of this work!
>
> 1. Is the specific reward design in Equation 2 strictly necessary? Specifically, how would the performance differ if final outcome correctness were used as the primary training label instead of the proposed consistency signals?
>
> In our ablation study on $\lambda$ in Section 4.9, we demonstrate that focusing more on self-consistency signal (larger $\lambda$) will lead to stronger optimization to step-wise accuracy, while focusing more on outcome signal will lead to stronger optimization to outcome accuracy. To demonstrate if the self-consistency part is necessary, we now conduct an additional experiment where $\lambda = 0$ in Alfworld setting. Specifically, we train over 100 steps and compare the final acc between $\lambda = 0$ and $\lambda = 1$ settings (the integrated feedback for policy keep unchanged, we only compare different signals for reward model). After optimization, we record the accuracy of process reward model as in the following table 2. We find the step acc drop significantly if we do not use self-consistency signal. We also find that the accuracies of policy on test set are 47.8 and 54.1, which demonstrates the self-consistency optimization for reward model can also benefit policy model's training.
>
> **Table 2**
> | $\lambda$ | Outcome Acc | Step Acc |
> |-|-|-|
> | 0 | 62.7 | 48.2 |
> | 1 | 61.0 | 54.7 |
>
>
> 2. The joint optimization of the policy and reward function resembles Actor-Critic frameworks, such as PPO. Have the authors performed a direct comparison with a PPO baseline, or could they more explicitly discuss the structural differences between these approaches?
>
> The co-optimization of the value model has a different purpose from our method. Training the value model is intended to improve the estimation of token-level advantage (the value model’s output), which is used to stabilize RL training. However, in our co-optimization approach for the reward model, the purpose is to optimize the reward model concurrently to improve policy model training. In fact, training the value model and the reward model are completely independent, so we can add value-model training to our approach. In the AlfWorld setting, we add an experiment using a value model. We find that using a value model makes no big difference compared to not using a value model in terms of policy optimization in our setting. Specifically, we find that the final policy accuracies are 54.1 without a value model and 54.4 with a value model after 100 steps of training.
>
> Please feel free to raise any further questions, and we will do our best to address them.

---

> > ### Author Rebuttal · Reviewer_PskM · 2026-04-06
> >
> > I think the authors addressed my concerns. I will maintain my score.

---

### Official Review · Reviewer_AsyS · 2026-03-17

**Soundness:** 3
**Presentation:** 3
**Significance:** 1
**Originality:** 1
**Overall Recommendation:** 2
**Confidence:** 4

**Summary:**

This paper presents RLAnything, a reinforcement learning framework where the policy, reward model, and environment tasks are all optimized together in a loop. The core idea is that the policy gets trained using both outcome rewards and step-wise feedback from a reward model (which the authors call integrated feedback), while the reward model itself gets improved during training through a consistency-based signal that checks whether its step-level judgments align with trajectory outcomes. On top of that, authors adapt the environment tasks automatically by using the reward model's evaluations as critic feedback to make tasks easier or harder depending on the policy's current ability level. They test across three pretty different settings: GUI computer use agents, text-based game agents, and coding LLMs.

**Compliance With Llm Reviewing Policy:**

Affirmed.

**Key Questions For Authors:**

1) How do you position your work relative to CRED and the broader UED and curriculum learning literature? CRED explicitly designs environments to improve reward model learning, which is conceptually very close to your claim that environment adaptation benefits reward model training.
2) Regarding the consistency feedback in Equation 2, since Rτi includes the reward model's own outputs, have you observed any degenerate behavior from this?

3) What is the actual computational cost of this framework compared to standard RLVR? You are running multiple reward model evaluations per step, doing environment adaptation with an additional LM call, and training the reward model alongside the policy. Some concrete numbers on wall clock time or FLOPs overhead would help readers judge the practical trade-off.

4) Can you provide a limitations section? For instance, when does environment adaptation fail to help, and are there task domains where the critic feedback produces poor or misleading task modifications?

**Limitations:**

no, authors should state clearly the limitations of their work.

**Strengths And Weaknesses:**

Starting with strengths, the paper tackles sparse rewards, a recurrent complex RL problem. The experimental coverage is good  (spanning GUI agents, text games, and coding tasks) which does shows generality. Authors include theoretical results connecting task difficulty balance to reward model precision (Theorems 3.1 and 3.2) which are the strongest contribution. The ablation studies are fairly thorough.

Now for weaknesses. My biggest concern is the missing discussion of highly relevant prior work, which makes the novelty claims feel overstated. The paper positions itself as if jointly optimizing environments for reward model learning is a new idea, but there is substantial prior work in curriculum learning, unsupervised environment design (UED), and active reward learning that are very close to what is discussed here.

On presentation, the paper is mostly well presented  but the framing question in the introduction ("If there exists an RL system that jointly optimizes...") needs review. Authors also missed being more upfront about limitations.

Overall, the empirical gains against the selected baselines are solid. But without proper contextualization against the curriculum learning and environment design literature, and their SOTA, this paper needs additional work.

Buening, Thomas Kleine, Victor Villin, and Christos Dimitrakakis. "Environment design for inverse reinforcement learning." arXiv preprint arXiv:2210.14972 (2022).
 Li, Mengdi, et al. "Curriculum-rlaif: Curriculum alignment with reinforcement learning from ai feedback." arXiv preprint arXiv:2505.20075 (2025).
Tung, Yi-Shiuan, Bradley Hayes, and Alessandro Roncone. "CRED: Counterfactual Reasoning and Environment Design for Active Preference Learning." 2026 IEEE International Conference on Robotics and Automation (ICRA). IEEE. 2026. (originally published July 2025).

---

> ### Author Rebuttal · Authors · 2026-03-31
>
> Thank you for taking the time to review our paper.
>
> 1. How do you position your work relative to CRED and the UED and curriculum learning literature. Missing discussion of relevant work.
>
> We appreciate the connection to CRED and the UED / curriculum-learning literature, regarding the part of changing the environment for better reward model optimization. In fact, we have already discussed UED and curriculum-learning works in related work section 2.2. We **will add the discussion of the three papers provided by the reviewer too**. However, our work studies **fundamentally different object**. We elaborate on the differences as follows:
>
> (1). **Main idea and objective:** RLAnything studies a dynamic RL system in which the policy, reward model, and environment are jointly optimized, so that each component strengthens the others through feedback. In contrast, the three cited works use environment only to improve the reward model, rather than the whole RL system. RLAnything’s novel environment adaptation improves both policy and reward-model training simultaneously through a dedicated reward system, supported by theoretical results. UED and curriculum-learning literature does not focus on the whole system, but on improving one aspect through data design.
>
> (2). **Methodology:** First, our reward-model learning is not preference learning like CRED and Curriculum-RLAIF. Our reward model receives scalar consistency feedback derived from outcome and PRM. Second, our environment adaptation mechanism is entirely different. Our environment is automatically adapted based on critic feedback of policy and reward model. CRED instead optimizes environment parameters to maximize the information gain of preference queries, while ED-IRL (Environment Design for Inverse Reinforcement Learning) selects environments to elicit more demonstrations. Third, our training loop is a coupled iterative loop to improve the whole system together. Finally, our reward system is also different: we explicitly combine process reward with verifiable outcome to achieve better optimization.
>
> (3). **Theory:** Our theory shows that balanced task difficulty keeps the loss estimator aligned with precision objective and serves as inspiration for method. By contrast, CRED and Curriculum-RLAIF are primarily methodological/empirical and not theorey-driven; ED-IRL includes a lemma which mainly justifies the objective, while our theory directly motivates method.
>
> (4). **Setting:** RLAnything focuses on LLM and agentic RL, including computer-use tasks, text-based games, and coding tasks. CRED is a robotics paper. ED-IRL focuses on maze and continuous-control settings (also no LLM). Curriculum-RLAIF is an LLM alignment paper in the standard RLAIF pipeline; it does not study complex interactive settings.
>
> 2. Rτi includes the reward model's own outputs, have you observed any degenerate behavior from this?
>
> To the best of our knowledge, we did not observe degeneration. The accuracy of the PRM improves promisingly after training (Table 1 in the paper), and can provide stronger training signal than verifiable outcome (figure 5). We also add the curve of reward model accuracy vs. training step in the AlfWorld setting in the following Table 1. We find that accuracy improves promisingly as training goes on. Rτi includes multiple evaluations by the reward model to achieve self-consistency optimization.
>
> **Table 1**
> | Step | Outcome Acc | Step Acc |
> |-|-|-|
> | 0 | 0.600 | 0.470 |
> | 50 | 0.620 | 0.530 |
> | 100 | 0.624 | 0.555 |
> | 150 | 0.625 | 0.562 |
> | 200 | 0.624 | 0.562 |
>
> 3. Actual computational cost compared to standard RLVR?
>
> Under the async RL framework, RLAnything incurs only slight additional overhead compared to standard RLVR: about 1.19× in AlfWorld and 1.25× in OSWorld, which is worthwhile given the significant gains in joint policy and reward-model improvement, as well as environment scaling. Once the next state is observed, the evaluation job is sent to the PRM asynchronously, so it does not interfere with the policy’s inference. Moreover, policy and reward model trainings are decoupled. However, our method does require additional GPUs to host the PRM. We will add this limitation.
>
>
> 4. Provide a limitations section?
>
> Thanks for the reminder; we will add a limitation section in the upcoming edition. First, we need additional GPUs (compared to standard RLVR) to host the PRM, but this is worthwhile since we optimize both the reward and policy models and achieve environment scaling in a closed loop. Second, the diversity of generated environment tasks may be limited by the ability of the model used to adapt tasks. A potential improvement is to use best-of-N and majority voting to select the best adaptation.
>
> Please feel free to raise any further questions, and we will do our best to address them.

---

> > ### Author Rebuttal · Reviewer_AsyS · 2026-04-01
> >
> > I want to thank the authors for their detailed response.
> >
> > Regarding the existing literature review in section 2.2. It is a very brief 2 paragraph pass through some related works. Adding a discussion as in the response would be very beneficial to the paper.
> >
> > Reading the authors response, I think my key concern is not addressed still. I never claimed this work is not novelty but rather how does it build on previous works.
> >
> > E.g.:
> >
> > "RLAnything studies a dynamic RL system in which the policy, reward model, and environment are jointly optimized, so that each component strengthens the others through feedback. In contrast, the three cited works use environment only to improve the reward model, rather than the whole RL system. RLAnything’s novel environment adaptation improves both policy and reward-model training simultaneously through a dedicated reward system"
> >
> > How does this "dedicated reward system" differs from imroving environment and reward model of other woeks? In RL improving the rewards is how we improve the policy. Reading from the paper you say:
> >
> > "During policy optimization, the policy’s trajectories also serve as the training environment for the reward
> > model. Meanwhile, the refined reward model provides a
> > stronger reward signal for the policy and more accurate feedback to guide environment task adaptation, which in turn
> > facilitates training of both the policy and the reward model." I still don't see this being fundamentally different from existing works.
> >
> > While your partcular methodology is novel I still see the whole work hindered because authors haven't provided any SOTA baselines, nor I find yet the theoretical discussion provides ground into how this is new methodology is better suited.
> >
> > I understand that it is now difficult to add baselines on the experimental section, but from the discussions alone so far my concerns remain

---

> > > ### Author Response · Authors · 2026-04-02
> > >
> > > Thank the reviewer for the fast response and appreciation of novelty of our work.
> > >
> > > 1. While methodology is novel, but haven't provided any SOTA baselines.
> > >
> > > We did not include baselines other than GRPO because no existing method is specifically designed to jointly improve the whole RL system like RLAnything. However, during rebuttal, we added comparisons with widely used UED and curriculum learning methods in the LLM community.
> > >
> > > (1). For the curriculum learning baseline, we train on tasks from easy to hard. Specifically, we first rank tasks in each OSWorld category based on the policy model’s accuracy, and divide them into four groups using thresholds of 1.0, 0.75, 0.5, and 0.25. We then train for 160 steps, spending 40 steps on each group. The final test accuracy is 34.6, substantially lower than the 37.5 achieved by our environment adaptation approach under the same 160-step budget.
> > >
> > > (2). For the UED baseline, we ask the environment model to adapt tasks based on the policy’s accuracy on raw tasks, like RLVE (https://arxiv.org/abs/2511.07317). Specifically, when a task’s accuracy is below 0.2 or above 0.8, we ask the environment model to make the task easier or harder, but without the critic feedback used in our method. The mechanism for accepting new tasks remains the same as in our method to ensure effective adaptation. We train for 160 steps on OSWorld and find that, without critic feedback, the final policy accuracy drops from 37.5 to 35.3. Further analysis shows that the number of accepted new tasks also decreases from 227 to 182. This demonstrates that critic feedback enables more effective environment adaptation by making it more directive and informative.
> > >
> > > These results demonstrates that RLAnything outperforms these most widely used method in LLM literature. We will add these two experiment results in our paper.
> > >
> > > 2. How does this "dedicated reward system" differs from imroving environment and reward model of other works? In RL improving the rewards is how we improve the policy. I don't see this being fundamentally different from existing works.
> > >
> > > (1). The existing works do not jointly optimize the entire RL system in the way RLAnything does, and such joint optimization is important. For example, reward-model-oriented curriculum learning methods such as Curriculum-RLAIF rely on an environment data collection and categorization process that requires human intervention. Although they can later train the policy model with the optimized reward model, this pipeline is limited in its level of automation and therefore cannot scale indefinitely. In contrast, our method improves the policy and reward models simultaneously without any human intervention, and can scale much more naturally, because the reward model is trained on trajectories generated by the policy model, which provides an effectively unlimited source of training data. Similarly, methods that use environment design to improve the policy model, such as RLVE, also require human effort to pre-design an environment engine that provides different difficulty levels for each task. In RLAnything, however, environment adaptation can be performed automatically based on critic feedback from both the reward model and the policy model. The superiority of RLAnything comes from its closed-loop optimization framework, in which each component provides feedback to the others without external intervention, enabling the system to scale continuously.
> > >
> > > (2). The current literature mainly focuses on designing environments to improve the training of either the policy model or the reward model, whereas the environment adaptation in RLAnything benefits both simultaneously. For example, RLVE adapts task difficulty to improve policy training, but it does not address improvement of the reward model, which could in turn further enhance policy learning. On the other hand, existing reward-model-oriented curriculum learning methods, including CRED, ED-IRL, and Curriculum-RLAIF, do not demonstrate that their environment adaptation improves policy training. In contrast, under our dedicated reward system, our environment adaptation strategy, which balances task difficulty, is shown both empirically and theoretically to improve the training of the policy model and the reward model at the same time. This dedicated reward system serves as the bridge between the two.
> > >
> > > (3). The reviewer states that “in RL, improving the reward is how we improve the policy.” However, this overlooks the fact that, in our framework, the policy is improved through three distinct mechanisms. First, the reward model is jointly optimized during training. Second, environment adaptation directly improves policy training by presenting tasks with better-calibrated difficulty. Third, environment adaptation also directly improves reward model training, which in turn further benefits policy optimization.
> > >
> > > Please feel free to raise any further questions, and we will do our best to address them.

---

### Decision · Program_Chairs · 2026-04-30

**Decision:**

Accept (regular)

**Comment:**

This paper proposes RLAnything, an RL framework in which the policy, reward model, and environment tasks are optimised together in a loop. Overall, the reviews are positive except for the one by the reviewer AsyS. The reviewer AsyS has a problem with the positioning of this work and the claim about what is novel. I do understand his concern based on the internal discussions. Given that the complaint is mostly about the positioning of the work in relation to other environment design and curriculum learning work, I recommend accepting the paper. I also strongly recommend that the authors establish these connections (as mentioned by reviewer AsyS) better in the final version of the paper. You do not need to compare with complex baselines in these adjacent fields, but it is valuable for readers to have a proper discussion of these related fields.